

# Protection without poison: Why tropical ozone maximizes in the interior of the atmosphere

Aaron Match[1], Edwin P. Gerber[1], and Stephan Fueglistaler[2]

[1]Center for Atmosphere Ocean Science, Courant Institute of Mathematical Sciences, New York University, New York, NY, USA

[2]Program in Atmospheric and Oceanic Sciences, and Department of Geosciences, Princeton University, Princeton, NJ, USA

**Correspondence:** Aaron Match (aaron.match@nyu.edu)

**Abstract.** Ozone is the most significant radiatively-active gas whose number density maximizes in the interior of the atmosphere, at an altitude of around 26 km in the tropics. Textbook explanations for this interior maximum begin by invoking the Chapman Cycle, a photochemical system that reproduces the altitude of maximum ozone despite omitting leading-order sinks from catalytic cycles and transport. Yet, these textbook explanations subsequently fragment into (1) a *source-controlled paradigm*, explaining ozone to maximize where its production rate maximizes, between abundant photons aloft and abundant $O_2$ below, and (2) a *source/sink competition paradigm* explaining ozone to maximize due to competition between the photolytic source and photolytic sink. Augmenting the Chapman Cycle with destruction by generalized catalytic cycles and transport, we demonstrate that these paradigms correspond to different regimes of ozone destruction, distinguished by whether photolysis of $O_3$ contributes at leading order to the sink. The tropical stratosphere is estimated to occupy a photolytic sink regime above 26 km and a non-photolytic sink regime below. Paradoxically, each paradigm predicts ozone to maximize outside its altitude range of applicability, motivating a new explanation, the *regime transition paradigm*: the interior maximum of ozone occurs at the transition from the photolytic sink regime aloft to the non-photolytic sink regime below. An explicit solution is derived for ozone under gray radiation, which produces an interior maximum at an endogenously-determined regime transition, and elucidates the ozone response to top-of-atmosphere UV perturbations.

## 1 Introduction

Ozone's presence in the stratosphere protects life from harmful UV radiation. It was the absence of this high-energy radiation at the surface that enabled Hartley to deduce the existence of the ozone layer (Hartley, 1881). In addition to protecting life from UV, ozone is also a strong oxidizing agent, making it poisonous to lungs and plant tissues. Thus, by maximizing well above the surface, around 30 km in the tropics, the ozone layer provides protection without poison.

The interior maximum of ozone distinguishes it from other radiatively-active atmospheric gases that are either well-mixed (e.g., carbon dioxide or methane) or thermodynamically confined near the surface (water vapor). This interior maximum of ozone is well-reproduced by state-of-the-art chemistry-climate models, which include complex representations of the atmospheric circulation and hundreds of chemical reactions. Here, we seek to drastically reduce the apparent complexity, and distill

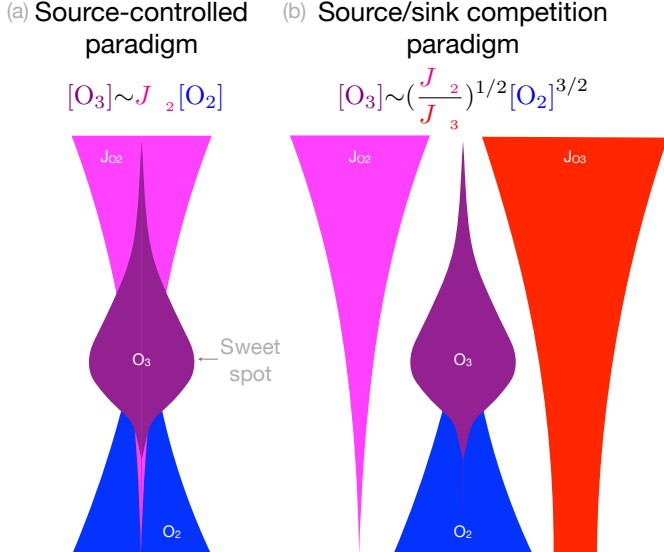

**Figure 1.** Two textbook paradigms for explaining the interior maximum of ozone. (a) In the source-controlled paradigm, ozone scales with its production rate given by the product of the photolysis rate ($J_{O_2}$), which decreases towards the surface, and the number density of [$O_2$], which increases exponentially towards the surface. Their product has been argued to maximize at a sweet spot. (b) In the source/sink competition paradigm, ozone scales as in the Chapman Cycle, with dependence on [$O_2$] and on the ratio of photolysis rates of $O_2$ and $O_3$. Only the source/sink competition paradigm invokes that photolysis of $O_3$ suppresses $O_3$.

the minimal set of physical and chemical processes required to explain the vertical structure of Earth's ozone profile, in partic-
ular the interior maximum of ozone number density around an altitude of 26 km in the tropics.

For almost a century, minimal explanations of the tropical ozone maximum have considered ultraviolet photochemistry, beginning with Chapman (1930), who showed how a motionless atmospheric column illuminated by ultraviolet light could produce an ozone layer through a photochemical cycle involving only species of oxygen, i.e., O, $O_2$, and $O_3$. We surveyed ten atmospheric radiation and chemistry resources (nine textbooks and one monograph[1], hereafter referred to as textbooks
for simplicity), and found that all ten explain the tropical ozone maximum in terms of the Chapman Cycle (Chapman, 1930). Yet, the textbooks then fragment into two explanatory paradigms. These paradigms have not been previously identified or compared, making it unclear whether they correspond to different orders of approximation or different physical assumptions. These paradigms predict different locations for the ozone maximum, exhibit different sensitivities to perturbations, and afford unequal status to the Chapman Cycle.
The first paradigm, invoked in 7 of the 10 sampled textbooks, is the *source-controlled paradigm*. It asserts that the interior maximum of ozone follows from the interior maximum in the ozone production rate (molec cm$^{-3}$ s$^{-1}$). The ozone production

---

[1]Monograph: Dutsch (1968); Textbooks: Jacob (1999); Liou (2002); McElroy (2002); Brasseur and Solomon (2005); Hites and Raff (2012); Calvert et al. (2015); Visconti (2016); Seinfeld and Pandis (2016); Brasseur and Jacob (2017)





rate is in turn argued to maximize in the interior of the atmosphere because it results from the product of the photolysis rate of $O_2$, $J_{O_2}$ ($s^{-1}$), which is large aloft but attenuates rapidly towards the surface, and the number density of $O_2$, denoted $[O_2]$ (molec $cm^{-3}$), which increases exponentially towards the surface. This product of $J_{O_2}$ and $[O_2]$ is understood to have an

interior maximum at a "sweet spot". This paradigm suggests that the interior maximum of ozone is a fundamental consequence of radiative attenuation through an exponentially-distributed absorber (Jacob, 1999), for which the radiative absorption rate (photons $cm^{-3}$ $s^{-1}$) maximizes where optical depth equals one. A cartoon version of this paradigm is shown in Fig. 1a.

The second paradigm, invoked in 3 of the 10 textbooks, is the *source/sink competition paradigm* (Fig. 1b). The source/sink competition paradigm uses the precise functional form of ozone derived from the Chapman Cycle. It describes the interior

maximum of ozone as following from the competition between the photolysis rate of $O_2$ ($J_{O_2}$, which when multiplied by $[O_2]$ gives the ozone production rate), and the photolysis rate of $O_3$ ($J_{O_3}$), with their ratio suitably weighted by some power of the number density of air, $n_a$ (molec $cm^{-3}$), which increases exponentially towards the surface.

These two paradigms make different assumptions about the ozone sink. By construction, the source-controlled paradigm neglects any structural contribution from the sink, known to be a tenuous assumption. Fig. 2 shows an assimilated ozone

profile from the atmospheric reanalysis MERRA-2, with peak number density around 26 km. A numerical calculation of the ozone profile with the Chapman Cycle reactions (see Section 2) reproduces the altitude of peak $O_3$, yet the Chapman Cycle source of ozone (blue curve) is known to be displaced roughly 20 km above the $O_3$ maximum. In general, the source would only be expected to align with ozone itself if the sink of ozone resembled passive relaxation (i.e., a term $\partial[O_3]/\partial t = -\kappa_{O_3}[O_3]$ with insignificant vertical structure in $\kappa_{O_3}$), so the Chapman Cycle appears to have an active sink. This Chapman sink results

from collisions of $O_3$ and $O$, the latter primarily produced by photolysis of $O_3$, so depends on the UV flux that is itself a function of the $O_3$ profile.

The second paradigm directly inherits the Chapman sink. Yet, the structural effects of this sink defy easy explanation, because the photolysis of $O_3$ depends on UV that is an implicit function of $O_3$ aloft. Worse, the Chapman Cycle sink accounts for less than 10% of the observed ozone sink, which is instead dominated by catalytic cycles and transport (e.g., Bates and

Nicolet, 1950; Crutzen, 1970; Jacob, 1999; Brasseur and Solomon, 2005). The missing sinks in the Chapman Cycle explain why it overestimates $O_3$ by approximately 50% (Fig. 2). So, the first paradigm invokes an unspecified passive sink that is inconsistent with the offset between the peak source and peak $O_3$, and the second paradigm invokes a specific yet minor sink.

We seek a minimal theory of the ozone maximum that invokes realistic sinks from catalytic cycles and transport. If these sinks are passive, then they might accord with the source-controlled paradigm. If active, then they may or may not accord with the

specific structural form of the Chapman Cycle from the source/sink competition paradigm. To represent these sinks, we bridge the gap between the Chapman Cycle and chemistry-climate models by augmenting the Chapman Cycle with two chemical reactions that represent the generalized destruction of $O$ and $O_3$ by catalytic cycles and transport. We call our approach the *Chapman+2 model*. The damping coefficients of the Chapman+2 model are constrained by the known magnitudes of catalytic cycles and transport.

Whether the damping is primarily of $O$ versus of $O_3$ will turn out to lead to qualitatively different mechanisms for ozone structure, a surprising result given that $O$ and $O_3$ are often treated as conceptually fungible within the chemical family of

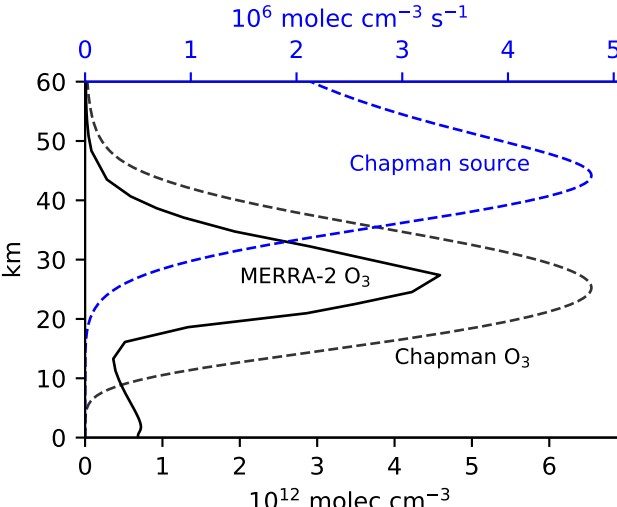

**Figure 2.** Profiles of $O_3$ and production rate of $O_3$. The bottom x-axis (black) shows ozone profiles in the Chapman Cycle photochemical equilibrium (dashed, described in methods) and the observed tropical average ozone profile from MERRA-2 (small differences in observational profiles are insignificant for the focus of this paper). The top x-axis (blue) shows the $O_3$ production rate (molec cm$^{-3}$ s$^{-1}$) in the Chapman Cycle.

odd oxygen ($O_x \equiv O + O_3$) (Section 2). In the O-damped limit, the destruction of ozone requires atomic oxygen that is supplied by photolysis of ozone, and therefore the ozone layer has a *photolytic sink*, which is an active sink analogous to that in the Chapman Cycle. In the $O_3$-damped limit, the sink does not rely on atomic oxygen produced by photolysis, and instead resembles the passive sink consistent with the source-controlled paradigm.

Today's tropical stratosphere occupies each regime at different altitudes, with the transition from a *photolytic sink regime* aloft to a *non-photolytic sink regime* below at 26 km, co-located with the ozone maximum (Section 3). Although each paradigm is capable of producing an interior maximum of ozone, neither can successfully explain the observed altitude of the tropical ozone maximum, which is instead best explained by a new mechanism: the *regime transition paradigm* (Section 4). In the regime transition paradigm, peak $O_3$ occurs at an altitude around 26 km precisely because this marks the transition from a photolytic sink regime aloft, within which ozone is increasing towards the surface, to a non-photolytic sink regime below, within which ozone is decreasing towards the surface. We present an analytical expression for an idealized ozone profile under gray radiative transfer that produces an interior maximum of ozone at an endogenously-determined regime transition, and which reproduces key sensitivities of the Chapman+2 model (Section 5).



## 2 The Chapman+2 model with destruction by generalized catalytic cycles and transport

A critical evaluation of the ozone maximum requires a model that can represent the structural effects of the ozone sinks from the Chapman Cycle, catalytic cycles, and transport. We briefly introduce the Chapman Cycle, which we then augment it with two generalized reactions to emulate the effects of catalytic cycles and transport. The Chapman Cycle reactions are:

$$\text{O}_2 + h\nu \quad \rightarrow \quad \text{O} + \text{O} \qquad (\lambda < 240 \text{ nm}) \tag{R1}$$

$$\text{O} + \text{O}_2 + M \quad \rightarrow \quad \text{O}_3 + M \tag{R2}$$

$$\text{O}_3 + h\nu \quad \rightarrow \quad \text{O}_2 + \text{O} \qquad (\lambda < 1080 \text{ nm}) \tag{R3}$$

$$\text{O} + \text{O}_3 \quad \rightarrow \quad 2\text{O}_2 \tag{R4}$$

Reactions R2 and R4 depend on collisions, where M is a third body whose number density is that of air ($n_a$). The combination reactions proceed as the number density of the chemical reactants multiplied by a rate coefficient $k_i$, i=2,4, e.g., reaction 2 has a rate of $k_2[\text{O}][\text{O}_2][M]$, which in general depends on temperature. Reactions R1 and R3 are photolysis reactions, and proceed as number density of the photolyzed species multiplied by the photolysis rate ($J_{\text{O}_2}$ or $J_{\text{O}_3}$). Photolysis rates couple chemistry and radiation together as follows:

$$J_{\text{O}_2}(z) = \int_\lambda q_{\text{O}_2}(\lambda)\sigma_{\text{O}_2}(\lambda)I(z,\lambda)d\lambda \tag{1}$$

$$J_{\text{O}_3}(z) = \int_\lambda q_{\text{O}_3}(\lambda)\sigma_{\text{O}_3}(\lambda)I(z,\lambda)d\lambda \tag{2}$$

with wavelength $\lambda$, quantum yield $q_i(\lambda)$ (molecules decomposed per photon absorbed by species $i$), absorption coefficient $\sigma_i$ (cm$^2$ molec$^{-1}$) (shown in Fig. 3b), and UV flux density with respect to wavelength $I(z,\lambda)$ (photons cm$^{-2}$ s$^{-1}$ nm$^{-1}$). Top-of atmosphere UV flux ($I(\infty,\lambda)$) is shown in Fig. 3a. Photolysis attenuates the UV flux:

$$I(z,\lambda) = I(\infty,\lambda)\exp(-\frac{\tau(z,\lambda)}{\cos\theta}) \tag{3}$$

where $\tau(z,\lambda)$ is the optical depth as a function of wavelength, and $\theta$ is the solar zenith angle, hereafter taken to be overhead sun for simplicity, so $\cos\theta = 1$. Because both O$_2$ and O$_3$ absorb UV, the optical depth at a given altitude depends on column-integrated O$_2$ and O$_3$ above that level:

$$\tau(z,\lambda) = \sigma_{\text{O}_2}(\lambda)\chi_{\text{O}_2}(z) + \sigma_{\text{O}_3}(\lambda)\chi_{\text{O}_3}(z) \tag{4}$$

where optical depth depends on the overhead column O$_2$ ($\chi_{\text{O}_2} = \int_z^\infty [\text{O}_2]dz$) and the overhead column O$_3$ ($\chi_{\text{O}_3} = \int_z^\infty [\text{O}_3]dz$).





## 2.1 Generalized destruction by catalytic chemistry and transport

The Chapman Cycle neglects catalytic chemistry and transport, but these processes are known to dominate the sink of ozone (Bates and Nicolet, 1950; Crutzen, 1970; Jacob, 1999; Brasseur and Solomon, 2005). We will incorporate generalized destruction from these processes as prescribed damping rates of O and $O_3$, with the damping rates constrained by observed magnitudes of specific catalytic cycles and transport.

For catalytic chemistry, we consider three general cycles of catalytic destruction by a catalyst Z, with each cycle distinguished

by its net effects. Representative cycles that lead to each net effect are shown in the following table:

| | |
|---|---|
| $Z + O_3$ | $\longrightarrow ZO + O_2$ |
| $ZO + O_3$ | $\longrightarrow Z + 2O_2$ |
| **Net:** $2O_3$ | $\longrightarrow 3O_2$ |
| $ZO + O$ | $\longrightarrow Z + O_2$ |
| $Z + O_2 + M$ | $\longrightarrow ZO_2 + M$ |
| $ZO_2 + O$ | $\longrightarrow ZO + O_2$ |
| **Net:** $2O$ | $\longrightarrow O_2$ |
| $Z + O_3$ | $\longrightarrow ZO + O_2$ |
| $ZO + O$ | $\longrightarrow Z + O_2$ |
| **Net:** $O + O_3$ | $\longrightarrow 2O_2$ |

The most significant catalysts driving each class of catalytic cycle are as follows (e.g., Brasseur and Solomon, 2005): destruction of $O_3$ is driven by Z = OH, destruction of O is driven by Z = H, and destruction of $O + O_3$ is driven by Z = H, OH, NO, Cl, and Br.

For transport, we focus on capturing the tropical lower stratosphere, which is known to be dominated by transport, with a balance between photochemical production and upward advection (Perliski et al., 1989; Brasseur and Jacob, 2017). There, tropical upwelling ($\bar{w}^*$) transports ozone-poor air from below, effectively damping ozone (Match and Gerber, 2022). Outside the tropical lower stratosphere, transport does not generally damp ozone, as is well understood by the role of transport in accumulating ozone in the mid-latitude lower stratosphere (e.g., Dobson, 1956). Although 1D treatments of tropical lower

stratospheric transport of $O_3$ have previously used an advective framework (e.g., Stolarski et al., 2014; Match and Gerber, 2022), here we will approximate its effects even more crudely as a damping in order to treat it commensurately with chemical sinks.

Thus, we propose that a minimal augmentation of the Chapman Cycle to include the effects of catalytic cycles and transport is to distill their effects into two generalized reactions that destroy and O and $O_3$:

$$O \quad \overset{z_O}{\rightarrow} \quad \frac{1}{2}O_2 \tag{R5}$$

$$O_3 \quad \overset{z_{O_3}}{\rightarrow} \quad \frac{3}{2}O_2 \tag{R6}$$




These generalized reactions represent two pathways for the destruction of odd oxygen: destruction of odd oxygen can scale with atomic oxygen, as in R5 that proceeds at the rate $\kappa_O[O]$, or it can scale with ozone, as in R6 that proceeds at the rate $\kappa_{O_3}[O_3]$. These reactions can be incorporated into the Chapman Cycle to yield a Chapman+2 model of tropical stratospheric ozone. When solving for the ozone profile in the Chapman+2 model, there is generally several orders of magnitude more $O_2$ than odd oxygen ($O_x \equiv O + O_3$), so for simplicity, $O_2$ will be treated as external to the Chapman+2 model, with fixed molar fraction of $C_{O_2} = 0.21$. Under this assumption, it is possible to solve for the number densities of O and $O_3$ in photochemical equilibrium by setting $\partial[O]/\partial t = \partial[O_3]/\partial t = 0$ in Reactions R1-R4, R5, and R6, and solving for this system of two equations in two variables (O and $O_3$):

$$[O_3] = \frac{J_{O_2} k_2}{k_4} C_{O_2}^2 n_a^3 \frac{1}{J_{O_3}[O_3] + J_{O_2} C_{O_2} n_a + \frac{\kappa_{O_3}[O_3]}{2} + \frac{J_{O_3} \kappa_O}{2k_4} + \frac{k_2 \kappa_{O_3} C_{O_2} n_a^2}{2k_4} + \frac{\kappa_O \kappa_{O_3}}{2k_4}} \tag{5}$$

where square brackets indicate number density (molec cm$^{-3}$), $n_a$ is the number density of air. This equation is quadratic in $[O_3]$ and mathematically implicit due to the dependence of $J_{O_2}$ and $J_{O_3}$ on ozone aloft. Note that $J_{O_3}$ appears in the denominator as a photolytic sink of ozone.

An accompanying diagnostic equation for atomic oxygen is as follows:

$$[O] = \frac{J_{O_2} C_{O_2} n_a + J_{O_3}[O_3] + \frac{\kappa_{O_3}[O_3]}{2}}{k_2 C_{O_2} n_a^2 + \frac{\kappa_O}{2}} \tag{6}$$

In the absence of catalytic cycles and transport, i.e., $\kappa_O = \kappa_{O_3} = 0$, Eqs. 5 and 6 reduce to the Chapman Cycle (e.g., as analyzed in Craig, 1965). The numerical details of our solution to Eqs. 5 and 6 are described in Appendix A.

## 2.2 Estimating generalized damping coefficients

We perform an order-of-magnitude estimate of the profile of dominant catalytic cycles and transport in order to estimate $\kappa_O$ and $\kappa_{O_3}$. We consider that the dominant catalytic regimes are the HO$_x$, NO$_x$, ClO$_x$, and BrO$_x$ cycles (e.g., Brasseur and Solomon, 2005). The HO$_x$ cycles can either destroy O or $O_3$, so we consider each term in the HO$_x$ cycle to contribute to damping of the relevant species. In the tropical stratosphere, the NO$_x$, ClO$_x$, and BrO$_x$ cycles destroy two O$_x$ (one O and one $O_3$), but are rate-limited by the destruction of O. This rate limitation arises because reaction Z + $O_3$ produces ZO, which can be photolyzed in a null cycle, such that only if ZO reacts with O does the catalytic cycle destroy two O$_x$. Therefore, such cycles are considered to damp odd oxygen at the rate of $2k_{ZO+O}[ZO]$. Catalytic reaction rates are taken from Brasseur and Solomon (2005).

For the effects of transport in the tropical lower stratosphere, we consider that transport damps both $O_3$ and O with a relaxation rate that scales with $\bar{w}^*$ divided by a reference vertical scale.

Combining the effects of catalytic cycles and transport, the effective catalytic damping rate of atomic oxygen, $\kappa_O$, is estimated as follows:

$$\kappa_O = \kappa_{\bar{w}^*} + a_5[OH] + a_7[HO_2] + 2b_3[NO_2] + 2d_3[ClO] + 2e_3[BrO] \tag{7}$$





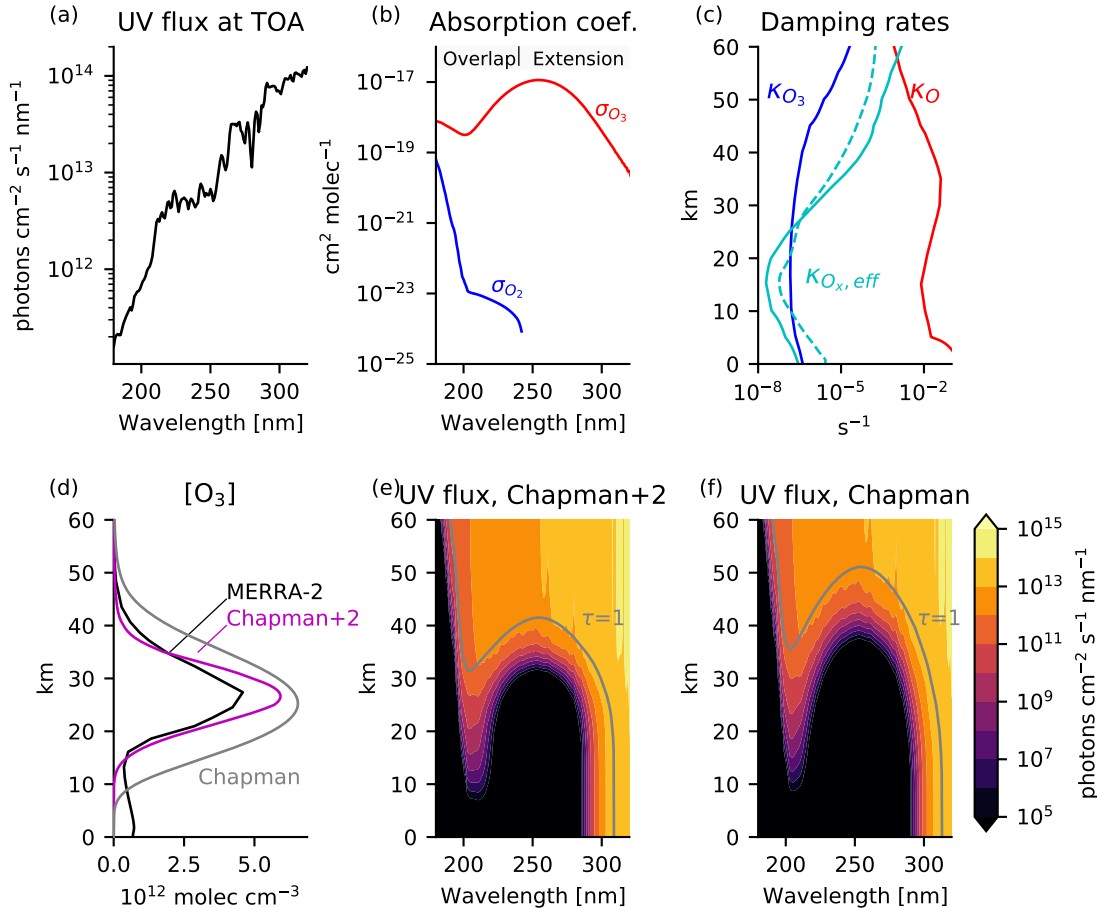

**Figure 3.** Boundary conditions and solutions to the Chapman Cycle and Chapman+2 model. (a) UV flux at the top of the atmosphere. (b) Absorption coefficients for $O_2$ and $O_3$. (c) Generalized damping rates of O (red) and $O_3$ (blue) estimated from Eqs. 7 and 8, using catalyst profiles from the chemistry-climate model SOCRATES, as tabulated in Brasseur and Solomon (2005). The effective damping rate of $O_x$ (solid cyan) is comparable to the derived $O_3$ relaxation rate in the chemistry-climate MOBIDIC as calculated in the Cariolle v2.9 linear ozone model (dashed cyan). (d) Ozone profile in numerical solutions to the Chapman Cycle. Numerical solutions are compared to ozone averaged from 30°S-30°N in 2018 in the Modern Era Retrospective analysis for Research and Applications Version 2 (MERRA-2), which blends direct observations with a state-of-the-art atmosphere model to provide a state estimate of the atmosphere (black). (e) UV flux for the damped case and (f) undamped case, indicating the level of unit optical depth ($\tau(\lambda) = 1$) in black. For clarity, wavelength axes are restricted to 180-320 nm although numerical solution extends up 800 nm into the weakly-absorbing Chappuis bands.

The effective damping rate of $O_3$ is:

$$\kappa_{O_3} = \kappa_{\bar{w}^*} + a_2[H] + a_6[OH] + a_{6b}[HO_2] \tag{8}$$



These damping rates are related to the budget of generalized odd oxygen ($O_y$) from Brasseur and Solomon (2005), because Eqs. 7 and 8 include all of the dominant sinks of $O_y$ that are linear in O or $O_3$. These damping rates neglect conversions of

generalized odd oxygen between reservoir species, so do not provide an exhaustive account of the $O_y$ budget. However, they serve our purpose of representing the damping of O and $O_3$ by catalytic cycles and transport.

Representative profiles for these damping rates are estimated by using globally-averaged vertical profiles for the chemical constituents from the chemistry-climate model SOCRATES (Brasseur et al., 1990), as tabulated in Brasseur and Solomon (2005). Our damping rates are approximated crudely insofar as profiles at the tropical latitudes of interest are being approxi-

mated by the globally-averaged profile. The damping by transport, is taken to be $\kappa_{\overline{w}^*} = (3 \text{ months})^{-1}$ (consistent with Fig. 5.3 in Brasseur and Solomon, 2005), corresponding to an ozone vertical scale of roughly 2 km.

The resulting profiles of effective damping rates are shown in Fig. 3c. These damping profiles can be compared to an independent estimate of the photochemical damping timescale, from the comprehensive chemistry-climate model MOBIDIC and linearized with respect to perturbations in ozone in the Cariolle v2.9 linear ozone model (Cariolle, 2023, personal communica-

tion, dashed cyan curve in Fig. 3c). These linear ozone model coefficients are equivalent to an effective damping of odd oxygen excluding the effects of transport, analogous in our framework to the quantity $\kappa_{O_{x,eff}} = (\kappa_{O_3} - \kappa_{\overline{w}^*}) + \gamma(\kappa_O - \kappa_{\overline{w}^*})$, where $\gamma \equiv [O]/[O_3]$ (solid cyan curve in Fig. 3c). These two cyan curves of the photochemical damping timescale are approximately consistent in magnitude and vertical structure, building confidence in our damping rates.

## 2.3 Validating the ozone profile

Fig. 3 compares numerical solutions of the Chapman Cycle and Chapman+2 model with the tropical ozone profile from MERRA-2. The Chapman Cycle is known to overestimate ozone by approximately 50% in the tropical stratosphere, as evident when comparing our Chapman solution to a representative tropical average ozone profile from MERRA-2 (Fig. 3d).

The overestimated ozone in the Chapman Cycle is partially corrected in the Chapman+2 model (Fig. 3d). The approximately corrected ozone magnitudes in the Chapman+2 model allow UV flux to penetrate more deeply than in the Chapman Cycle (Fig.

3e,f). Of course, agreement between the Chapman+2 model and MERRA-2 remains imperfect, with many possible sources for this discrepancy, including the neglect of diurnal and seasonal cycles in solar zenith angle, the approximation of transport as a linear damping, the use of globally-averaged (instead of tropically-averaged) catalytic profiles, and the neglect of vertical temperature variations. Despite these assumptions, the Chapman+2 model produces a reasonable fit to the observed profile, and will be considered to produce a credible interior maximum of ozone. The remainder of the paper seeks to explain why the

Chapman+2 model produces an interior maximum.

## 3 Understanding the ozone maximum: photolytic regimes in the tropical stratosphere

Understanding how the Chapman+2 model produces an interior maximum is challenging when considering the generalized ozone number density in Eq. 5, so we perform a scale analysis to identify limiting cases of the photochemical mechanisms, which correspond to photolytic regimes at different altitudes. Three pathways emerge for the sink of odd oxygen: the Chapman





Cycle sink from the reaction O + O$_3$, damping of O, and damping of O$_3$. Each pathway can separately dominate the destruction of ozone, corresponding to different dominant terms in the six-term denominator of Eq. 5.

    If the Chapman Cycle sink of ozone dominates, then the dominant term in the denominator of Eq. 5 is $J_{O_3}[O_3]$, and ozone scales as:

$$[O_3] = (\frac{J_{O_2} k_2}{J_{O_3} k_4})^{1/2} C_{O_2} n_a^{3/2} \tag{9}$$

Eq. 9 reproduces the well-known Chapman Cycle limit and is presented in most textbook explanations for the shape of the ozone layer. The vertical structure of the ozone layer in Eq. 9 arises from the number density of air, $n_a^{3/2}$ (assumed invariant to Chapman dynamics), and from the ratio of photolysis rates $(J_{O_2}/J_{O_3})^{1/2}$. The presence of $J_{O_3}$ in the denominator indicates that photolysis of O$_3$ is an effective sink of O$_3$ by producing atomic oxygen that can then destroy O$_3$ through R4. We refer to this as a *photolytic sink*. The fact that photolysis of O$_3$ acts as a photolytic sink might seem surprising since it is typically understood

to not affect ozone due to the strong null cycle between Reactions R2 and R3. However, that null cycle has some leakage into R4. Thus, even though most of the photolysis of ozone does not lead to the destruction of ozone (legitimating the concept of odd oxygen), most of the destruction of ozone requires photolysis of ozone, in order to supply atomic oxygen—hence $J_{O_3}$ suppresses ozone as a photolytic sink.

    If the damping of O dominates through $J_{O_3} \kappa_O / 2 k_4$, ozone number density scales as:

$$[O_3]_{\text{O-damped}} = \frac{2 J_{O_2} k_2 C_{O_2}^2 n_a^3}{J_{O_3} \kappa_O} \tag{10}$$

    The Chapman Cycle and O-damped limit (Eqs. 9 and 10) share key structural aspects, as ozone scales as $((J_{O_2}/J_{O_3}) n_a^3)^n$, where $n = 1/2$ in the Chapman Cycle, and $n = 1$ in the O-damped limit. Note that in both cases, photolysis of O$_3$ appears in the denominator as a photolytic sink that is necessary for producing atomic oxygen that can either react with ozone (in the Chapman Cycle) or be damped (in the O-damped limit). Thus, these limits are both *photolytic sink regimes*. Both satisfy the

source/sink competition paradigm.

    If the damping of O$_3$ dominates through $k_2 \kappa_{O_3} C_{O_2} n_a^2 / 2 k_4$, ozone number density scales as:

$$[O_3]_{\text{O}_3\text{-damped}} = \frac{2 J_{O_2} [O_2]}{\kappa_{O_3}} \tag{11}$$

    In the O$_3$-damped limit, ozone destruction does not depend on photolysis of ozone, which therefore does not appear in the ozone equation. This O$_3$-damped limit is therefore in a *non-photolytic sink regime*. With the passive sink of the non-photolytic

sink regime, ozone scales with the production rate divided by the damping rate of O$_3$, consistent with the source-controlled paradigm.

    Thus, the prevailing textbook paradigms for explaining the interior maximum of ozone correspond to well-defined limits of a Chapman+2 model that is either in the Chapman Cycle limit or O-damped limit (source/sink competition paradigm, Fig. 1b)



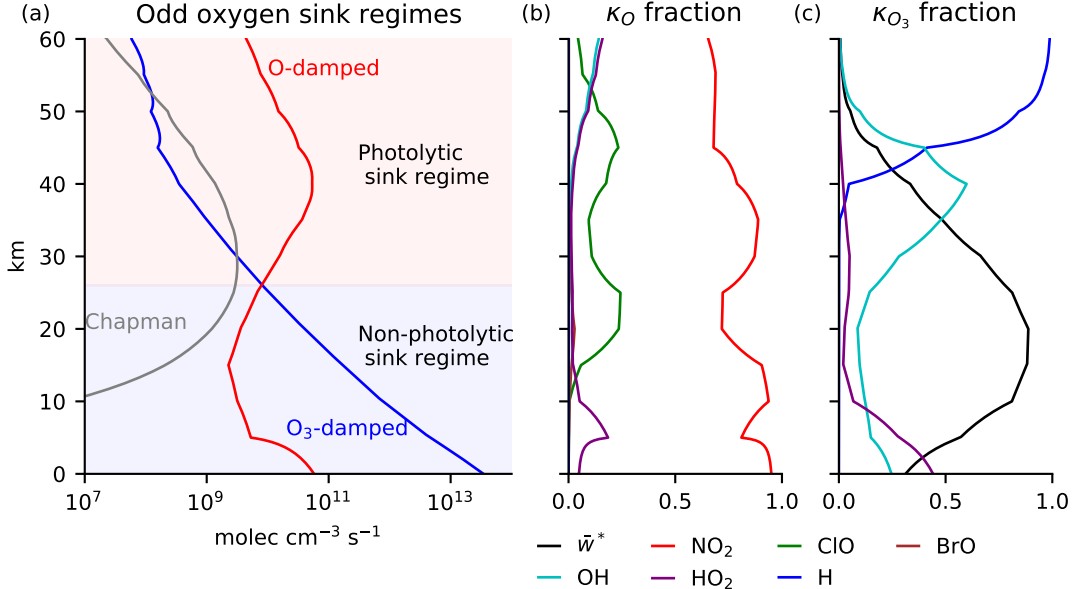

**Figure 4.** (a) Odd oxygen sink regimes due to catalytic chemistry and transport based on the effective damping rates of O and $O_3$. If the Chapman or O-damped terms are leading order, then the ozone layer is in a photolytic sink regime (red region above 26 km). If damping of $O_3$ is leading order, the ozone layer is in a non-photolytic sink regime (blue region below 26 km). (b) Fraction of $\kappa_O$ contributed by each component in Eq. 7. (c) Fraction of $\kappa_{O_3}$ contributed by each component in Eq. 8.

or in the $O_3$-damped limit (source-controlled paradigm, Fig. 1a). Which limit actually prevails is an empirical problem. Fig.
3c reveals that the damping of O is everywhere larger than the damping of $O_3$, but this does not imply that the ozone layer is everywhere in an O-damped regime. Instead, determining the O-damped versus $O_3$-damped limits requires evaluating the dominant terms in the denominator of the generalized catalytic ozone solution (Eq. 5), where the contribution of the Chapman Cycle sink scales as $J_{O_3}[O_3]$, the damping of O scales as $J_{O_3}\kappa_O/2k_4$, and damping of $O_3$ scales as $\frac{1}{2}\kappa_{O_3}([O_3]+k_2 C_{O_2} n_a^2/k_4)$.

The resultant catalytic regimes are shown in Figure 4a, and can be categorized as follows:

– The upper stratosphere is dominated by O-damping and is therefore in a photolytic sink regime down to 26 km. This means that an interior maximum of ozone occurring well above 26 km would be explainable within the source/sink competition paradigm.

– Below 26 km, damping of $O_3$ is dominant, leading to a non-photolytic sink regime. This means that an interior maximum of ozone well below 26 km would be explainable within the source-controlled paradigm.

Yet, he interior maximum of ozone in the Chapman+2 model occurs exactly at this transition, at an altitude of 26 km, hinting at the need to consider both regimes to explain tropical ozone on Earth.





The damping of O and $O_3$ that establishes each regime can be further decomposed into additive contributions from the terms in Eqs. 7 and 8 (Figs. 4b and 4c). Throughout the entire stratosphere, the damping of O is dominated by $NO_2$ (Fig. 4b, red curve). Thus, to leading order, $\kappa_O$ can be approximated as $b_3[NO_2]$. The damping of $O_3$ is dominated by H in the

upper stratosphere, by OH lower down around 40 km, and by transport below 35 km. The dominance of transport in the lower stratosphere means that the non-photolytic sink regime below 26 km is established primarily by $\kappa_{\overline{w}^*}$ (Fig. 4c, black curve). Thus, the odd oxygen sink regimes regimes can be approximated as a $NO_x$-driven O-damped regime above 26 km and a transport-driven $O_3$-damped regime below 26 km.

## 4   Ozone on our planet: a new theory for the observed ozone maximum

To examine whether the interior maximum of $O_3$ can be explained in terms of only the photolytic or non-photolytic sink regimes in isolation, we consider the predictions from each regime both inside and outside their altitudes of applicability. Fig. 5 shows the MERRA-2 ozone profile (black) compared to the Chapman+2 model solution (magenta) and its limits in the photolytic sink regime (solid red) and non-photolytic sink regime (solid blue). Above 26 km, the ozone number density closely follows the scaling of the photolytic sink regime (solid red). Below 26 km, the ozone number density closely follows

the scaling of the non-photolytic sink regime (solid blue) until reaching the tropopause, below which our model assumptions are no longer valid.

To examine where each regime predicts peak $O_3$, these theoretical scalings can be artificially extended beyond where they formally apply (dashed curves). When the photolytic sink regime is extended downwards (red dashed), it predicts an interior maximum at 17 km, far below the ozone maximum and far below the range of applicability of the photolytic sink regime.

When the non-photolytic sink regime is extended upwards (blue dashed), it predicts an interior maximum at 35 km, far above the ozone maximum and far above the applicability of the non-photolytic sink regime. Thus, a paradox has emerged: each textbook paradigm predicts an interior maximum at the wrong altitude and in a region where it does not apply.

A new mechanism must be responsible for the interior maximum of ozone. We propose that the tropical maximum in ozone number density occurs around 26 km precisely *because* this marks the transition from a photolytic sink regime aloft, within

which ozone is increasing towards the surface, to a non-photolytic sink regime below, within which ozone is decreasing towards the surface. We call this new mechanism the *regime transition paradigm*. The regime transition paradigm is illustrated in Fig. 6.

The existence of a regime transition from a photochemically-dominated regime to a transport-dominated regime around 25-30 km has been previously noted (Perliski et al., 1989; Brasseur and Jacob, 2017). However, we are not aware of a previous link

between peak ozone and the top of the transport-dominated regime. Furthermore, although the $O_3$-damped limit is presently dominated by transport, a regime transition from an O-damped limit to an $O_3$-damped limit can in principle occur due to catalytic cycles alone, even in a motionless atmosphere. The interior maximum of $O_3$ results from the regime transition, regardless of its cause.



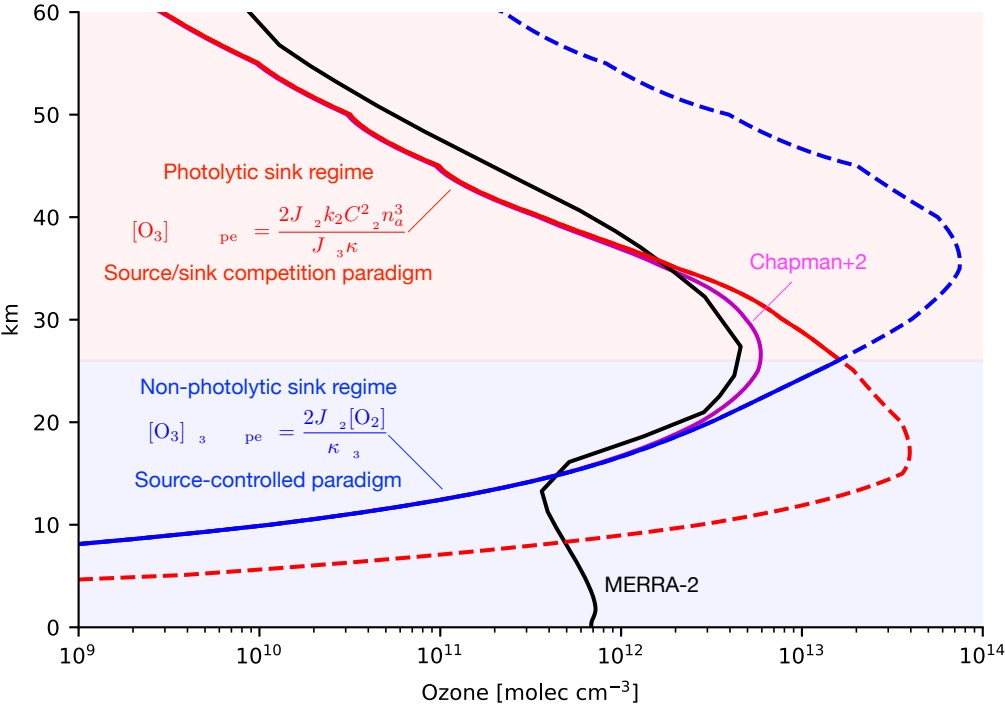

**Figure 5.** The Chapman+2 model (magenta) compared to a representative tropical ozone profile from MERRA-2 in 2018 (black). Above 26 km, ozone is in a photolytic sink regime (red shading), and follows the theoretical scaling for ozone in an O-damped photolytic sink regime (Eq. 10, i.e., $[O_3] = 2J_{O_2}k_2C_{O_2}^2n_a^3/J_{O_3}\kappa_O$; red). Below 26 km, ozone is in a non-photolytic sink regime (blue shading), and follows the theoretical scaling for ozone in an $O_3$-damped nonphotolytic sink regime (Eq. 11, i.e., $[O_3] = 2J_{O_2}[O_2]/\kappa_{O_3}$; blue). Extending the theoretical scalings across the whole domain (dashed curves) reveals the apparent paradox that each scaling predicts ozone to maximize its region of applicability. This reveals that the observed maximum results as ozone transitions from increasing towards the surface within the photolytic sink regime to decreasing towards the surface within the non-photolytic sink regime.

## 5   An explicit solution to the ozone maximum in a gray atmosphere

There are no mathematically explicit solutions to the ozone layer. This is due to two key obstacles: (1) ozone photochemistry is mathematically implicit, and (2) it relies on spectral integrals across non-analytic functions. The obstacle from spectral integrals across non-analytic functions is generic to radiative transfer problems. Yet, recent work has advanced understanding of the emergent effects of longwave radiative transfer by judiciously approximating non-analytic absorption spectra for $CO_2$ or $H_2O$ with analytic functions, leading to *simple spectral models* (SSMs, after Jeevanjee and Fueglistaler, 2020) that can

then be coupled to other aspects of climate dynamics (Jeevanjee and Fueglistaler, 2020; Jeevanjee et al., 2021; Pierrehumbert, 2011; Romps et al., 2022). Here, we develop simple spectral models for ozone photochemistry, in certain limits of which the



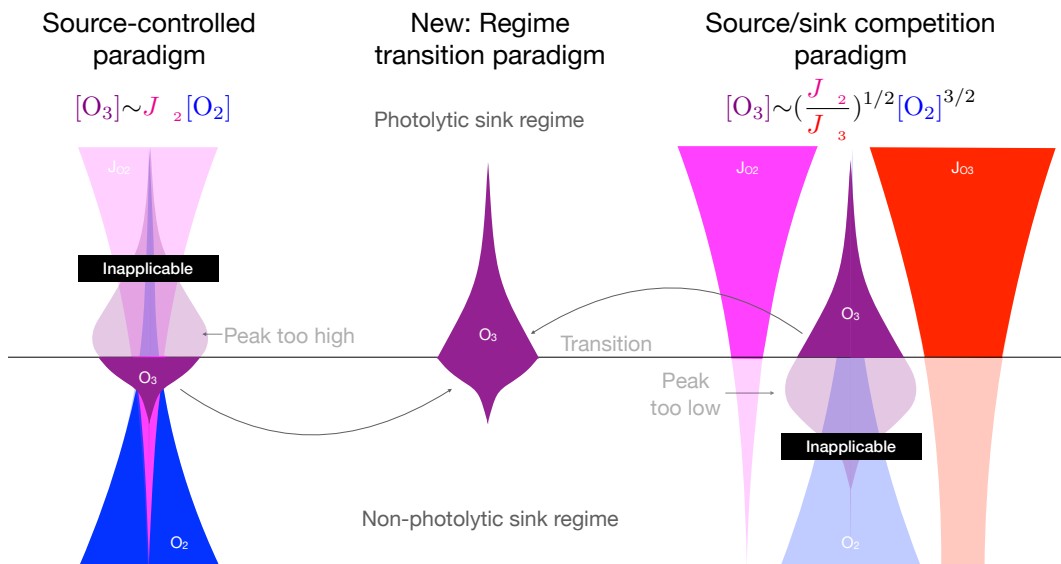

**Figure 6.** The regime transition paradigm (center) is proposed to explain the interior maximum of ozone after finding that the source-controlled paradigm (left) and source/sink competition paradigm (right predict an interior maximum outside their range of applicability, whereas the observed maximum occurs at the transition between these two regimes.)

UV radiative transfer becomes mathematically explicit. Under these approximations, we derive explicit theories for the ozone profile in all three paradigms.

The strictest simplification of spectral quantities is to consider gray radiative transfer, in which $O_2$ and $O_3$ are approximated as having spectrally-uniform absorption coefficients and quantum yields (assumed to be unity). Some previous texts have built intuition for the ozone layer using reasoning based on gray radiative transfer. In particular, Jacob (1999) has a textbook problem in Chapter 10 that seeks to explain the shape of the ozone layer by considering the photon absorption rate (molec $\text{cm}^{-3} \text{ s}^{-1}$) for a monochromatic beam of radiation in a spectral window $\mu$ with top-of-atmosphere flux of $I_\infty$ (photons $\text{cm}^{-2}$ $\text{s}^{-1} \text{ nm}^{-1}$) propagating from overhead into exponentially-distributed $O_2$ with absorption coefficient $\sigma^*_{O_2}$ ($\text{cm}^2 \text{ molec}^{-1}$). The ozone production rate in this case, equal to $J_{O_2}[O_2]$, is as follows:

$$J_{O_2}[O_2] = \sigma^*_{O_2} I_\infty \mu C_{O_2} n_{a_0} \exp(-\sigma^*_{O_2}\chi_{O_2}(z) - \frac{z}{H}) \tag{12}$$

As an expression for the ozone production rate, Eq. 12 can describe an interior maximum of ozone in the source-controlled paradigm. But, Eq. 12 describes neither the source/sink competition paradigm (including the Chapman Cycle) nor does it describe the more-realistic regime transition paradigm (Section 4). Furthermore, this expression neglects absorption of UV by ozone, and thus implicitly neglects UV protection by ozone or the effects it would have on the vertical profile of ozone.

We seek to develop an analytical theory for the ozone profile in the limit of gray radiation that accounts for absorption by ozone and that can produce an interior maximum at the transition from an O-damped regime aloft to an $O_3$-damped regime





below. To facilitate this analysis, we assume an isothermal atmosphere and uniform damping of O and $O_3$. Using these idealized boundary conditions, we present the first explicit solutions to an ozone layer.

We solve for ozone from the top down, beginning in a photolytic sink regime at high altitudes, where the UV flux is large enough to drive fast photochemical equilibration compared to the damping of $O_3$. Descending, the photochemical equilibration becomes more sluggish until reaching a transition altitude to an $O_3$-damped regime. In today's atmosphere (and across a quite wide parameter regime), this transition altitude is also the altitude of the ozone maximum. Below the transition altitude, the ozone layer is in the non-photolytic sink regime.

### 5.1    Upper branch in photolytic sink regime

The photolytic sink regime occurs wherever damping of O is stronger than damping of $O_3$. We consider the ozone profile that results under uniform and fully overlapping absorption across a window of spectral width $\mu$ with absorption coefficients $\sigma^*_{O_2}$ and $\sigma^*_{O_3}$. The photolysis rates can be expressed implicitly as a function of ozone by substituting into Eqs. 1 and 2:

$$J_{O_2}(z) = \mu \sigma^*_{O_2} q_{O_2} I_\infty \exp(-\sigma^*_{O_2} \chi_{O_2}(z) - \sigma^*_{O_3} \chi_{O_3}(z)) \qquad (13)$$

$$J_{O_3}(z) = \mu \sigma^*_{O_3} q_{O_3} I_\infty \exp(-\sigma^*_{O_2} \chi_{O_2}(z) - \sigma^*_{O_3} \chi_{O_3}(z)) \qquad (14)$$

The quantum yields are assumed to be unity and will be henceforth omitted from our simple spectral models because, for our purposes, their structure is redundantly encoded in the absorption coefficients. The photolysis rates depend on column ozone, so it would seem that the ozone profile should depend implicitly on ozone aloft. However, ozone in the photolytic sink regime scales with the ratio between the photolytic source and photolytic sink, which under gray radiation simplifies to the ratio of the absorption coefficients, i.e., $J_{O_2}/J_{O_3} = \sigma^*_{O_2}/\sigma^*_{O_3}$ (gray radiation only). This simplifies the expression for ozone in the photolytic sink regime and facilitates an explicit solution for ozone in the O-damped regime (Eq. 10):

$$[O_3]_{\text{gray,O-damped}} = \frac{2\sigma^*_{O_2} k_2 C^2_{O_2} n_a^3}{\sigma^*_{O_3} \kappa_O} \qquad (15)$$

In the gray O-damped limit, ozone number density increases proportionally to $n_a^3$. Absent a transition to a non-photolytic sink regime, the gray O-damped ozone layer would increase all the way down to a surface maximum! Thus, in general, a gray atmosphere in the photolytic sink regime cannot reproduce an interior maximum of ozone. Explanations of the interior maximum of ozone in the source/sink competition paradigm (including when applied to the Chapman Cycle) must invoke spectral structure, e.g., as in when Dutsch (1968) wrote the following: "The formation of a layer of maximum ozone content arises from the fact that below about 35 km the dissociation rate of molecular oxygen ($J_{O_2}$) drops off much more rapidly than that of ozone ($J_{O_3}$), mainly because of the overlap of ozone and oxygen absorption around 2,100 A (210 nm)."

Eq. 15 can be integrated to yield column ozone:



$$\chi_{O_3}(z)|_{\text{gray,O-damped}} = \frac{H}{3}[O_3]_{\text{gray,O-damped}} \tag{16}$$

This expression for the column ozone under monochromatic radiation and O-damping can then be substituted back into the photolysis rates to solve explicitly for $J_{O_2}$ (Eq. 13) and $J_{O_3}$ (Eq. 14).

Because the Chapman Cycle is also in the photolytic sink regime, these results can be adapted to yield analytical solutions
to a gray Chapman Cycle. As in the O-damped limit, the gray Chapman Cycle ozone layer has a surface maximum of ozone, both in terms of number density and molar fraction. Thus, the interior maximum of the Chapman Cycle cannot be explained by gray radiation arguments, or any argument that lacks an explicit spectral dimension. We consider explicit analytical solutions with simple spectral models to the Chapman Cycle in Appendix B, demonstrating how an interior maximum of ozone requires absorption by ozone in the extension window (an absorption feature noted in Fig. 3b).

## 5.2    Regime transition and peak $O_3$

The regime transition occurs where the damping of O and $O_3$ are exactly co-dominant, as determined by the terms in the denominator of Eq. 5. Damping of O scales with $J_{O_3}\kappa_O/2k_4$, and damping of $O_3$ scales with $k_2\kappa_{O_3}C_{O_2}n_a^2/2k_4$, so the equality of these two regimes leads to the following condition on $\kappa_{O_3}$ at the transition altitude, $z_t$:

$$\kappa_{O_3} = \frac{J_{O_3}(z_t)\kappa_O}{k_2 C_{O_2} n_a^2(z_t)} \tag{17}$$

To solve analytically for the height at which this condition is satisfied, it is necessary to make an assumption about the dominant absorber of UV, which we realistically take to be $O_3$. Under that assumption and using the column ozone scaling for the O-damped regime, the ozone photolysis rate scales as follows:

$$J_{O_3}(z) = \sigma^*_{O_3} I_\infty \Delta\lambda \exp(-\sigma^*_{O_3}\chi_{O_3}(z)|_{\text{gray,O-damped}}) \tag{18}$$

Substituting this expression for the photolysis rate of ozone into the transition condition (Eq. 17) and solving for $z$ yields the
transition altitude:

$$z_t = H\left(\frac{1}{3}W\left(\frac{\tau_{O_2}(0)\alpha_O^{1/2}}{\alpha_{O_3}^{3/2}}\right) + \frac{1}{2}\ln\frac{\alpha_{O_3}}{\alpha_O}\right) \tag{19}$$

where $W$ is the Lambert W function, which when evaluated at $x$ returns the value $w$ such that $w\exp(w) = x$, and we have defined the following three non-dimensional parameters of use for interpreting the transition altitude scaling:

$$\alpha_O \equiv \frac{\kappa_O}{k_2 C_{O_2} n_{a_0}^2} \tag{20}$$



$$\alpha_{O_3} \equiv \frac{\kappa_{O_3}}{\sigma_{O_3}^* I_\infty \Delta\lambda} \tag{21}$$

$$\tau_{O_2}(0) = \sigma_{O_2}^* C_{O_2} n_{a_0} H \tag{22}$$

The first nondimensional parameter, $\alpha_O$, measures the strength of O-damping compared to the rate at which atomic oxygen combines with $O_2$ to form $O_3$ (R2). The second nondimensional parameter, $\alpha_{O_3}$, measures the strength of $O_3$-damping compared to the rate at which $O_3$ is photolyzed at the top of the atmosphere (R3). The third nondimensional parameter is the optical depth of $O_2$ at the surface.

Substituting the expression for $z_t$ into the scaling for ozone in the O-damped regime (Eq. 15) yields an analytical expression for ozone at the transition altitude:

$$[O_3](z_t) = \frac{2}{H\sigma_{O_3}^*} W\left(\frac{\alpha_O^{1/2} \tau_{O_2}(0)}{\alpha_{O_3}^{3/2}}\right) \tag{23}$$

This is an explicit analytical expression for $O_3$ at the transition altitude, which for realistic parameters is also the peak $O_3$. Some of the dependencies in this expression are consistent with prior understanding, but others are surprising. For example:

– Increasing the absorption coefficient of $O_2$, $\sigma_{O_2}^*$, leads to an increase in peak $O_3$, because it increases the $O_3$ production rate to enhance $O_3$ everywhere.

– Increasing the ozone damping, $\kappa_{O_3}$, reduces the peak $O_3$.

– Surprisingly, increasing the damping of O *increases* peak $O_3$ despite reducing $O_3$ at any given altitude in the photolytic sink regime, because it also lowers the transition altitude.

– Increasing the incoming UV radiation increases peak $O_3$.

The sensitivity to an increase in UV will be worked in more detail and compared among analytical solutions to the paradigms in the Discussion. For now, we proceed with our derivation of the ozone profile into the non-photolytic sink regime.

### 5.3 Lower branch in non-photolytic sink regime

To solve for the lower branch of the ozone profile, we use the constraint that UV flux is continuous across the regime transition. Thus, our approach to solving for ozone in the non-photolytic sink regime is to consider an $O_3$-damped region below $z_t$ with constant $\kappa_{O_3}$.

In the non-photolytic sink regime, ozone scales with its photolytic production rate, $J_{O_2}$, which we solve for by substituting the expression for $O_3$ in the $O_3$-damped limit (Eq. 11) into the column ozone integral:





$$J_{O_2}(z) = \mu \sigma_{O_2}^* I_\infty \exp(-\sigma_{O_2}^* \chi_{O_2}(z)) \exp(-\sigma_{O_3}^* \int_z^{z_t} \frac{2 J_{O_2}[O_2]}{\kappa_{O_3}} dz) \tag{24}$$

Taking the natural logarithm of both sides of Eq. 24 and differentiating with respect to $z$ leads to a differential equation for $J_{O_2}$ as a function of $z$:

$$\frac{dJ_{O_2}(z)}{dz} = \frac{2 \sigma_{O_3}^* C_{O_2} n_{a_0}}{\kappa_{O_3}} J_{O_2}(z)^2 \exp(-z/H) + \sigma_{O_2}^* C_{O_2} n_{a_0} J_{O_2}(z) \exp(-z/H) \tag{25}$$

This first-order nonlinear ordinary differential equation can be solved by separation of variables and integrated from the

transition altitude $z_t$ downwards using the following boundary condition:

$$J_{O_2}(z_t) = \sigma_{O_2}^* \mu I_\infty \exp(-\sigma_{O_2}^* \chi_{O_2}(z_t)) \exp(-\sigma_{O_3}^* \chi_{O_3}(z)|_{\text{gray,O-damped}}) \tag{26}$$

which leads to an equation for $J_{O_2}$:

$$J_{O_2}(z) = \frac{\sigma_{O_2}^* \kappa_{O_3}}{2 \sigma_{O_3}^* ((\frac{\sigma_{O_2}^* \kappa_{O_3}}{2 J_{O_2}(z_t) \sigma_{O_3}^*} + 1) \exp(\tau_{O_2}(0)(e^{-z/H} - e^{-z_t/H})) - 1)} \tag{27}$$

This expression for $J_{O_2}(z)$ can be substituted into the equation for $O_3$ under $O_3$-damping to yield a profile of ozone in the

non-photolytic sink regime.

In Appendix B, we present an analytical solution to the gray ozone layer under strong damping such that $z_t$ can be approximated as the top of the atmosphere. This solution can reproduce the "sweet spot" explanation for the ozone layer (Eq. 12), of pedagogical and historical interest despite not emulating the observed tropical ozone profile.

### 5.4 Putting the pieces together

Summarizing, $O_3$ in the upper branch is in the photolytic sink regime (Eq. 15) down to the altitude of the regime transition, $z_t$ (Eq. 19). Below $z_t$, $O_3$ in the lower branch is in the non-photolytic sink regime (inferred from Eq. 27). Piecing these regimes together yields an explicit analytical profile of ozone in the gray Chapman+2 model:

$$[O_3]_{\text{gray}} = \begin{cases} \frac{2 \sigma_{O_2}^* k_2 C_{O_2}^2 n_{a_0}^3 \exp(\frac{-3z}{H})}{\sigma_{O_3}^* \kappa_O} & \text{if } z \geq z_t \\ \frac{2}{H \sigma_{O_3}^*} W(\frac{\alpha_O^{1/2} \tau_{O_2}(0)}{\alpha_{O_3}^{3/2}}) & \text{if } z = z_t \\ \frac{\sigma_{O_2}^* C_{O_2} n_{a_0} \exp(-z/H)}{\sigma_{O_3}^* ((\frac{\sigma_{O_2}^* \kappa_{O_3}}{2 J_{O_2}(z_t) \sigma_{O_3}^*} + 1) \exp(\tau_{O_2}(0)(e^{-z/H} - e^{-z_t/H})) - 1)} & \text{if } z < z_t \end{cases} \tag{28}$$



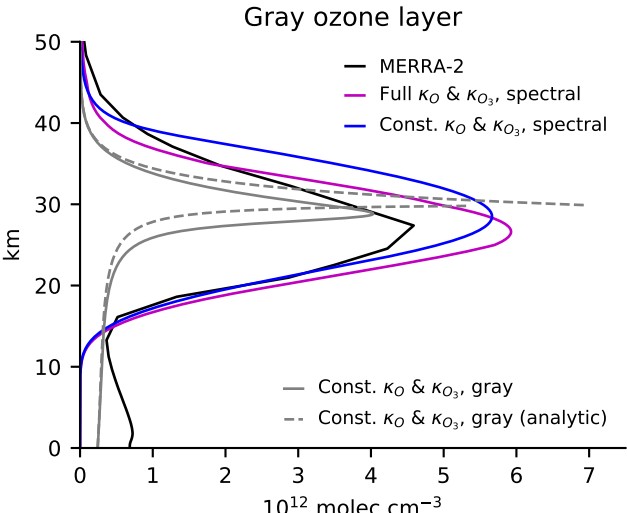

**Figure 7.** An explicit analytical solution to the ozone layer under gray radiation is compared at various levels of approximation. We show the assimilated ozone profile in MERRA-2 (black), the full damping solution with spectral radiation (magenta, identical to magenta curve in Fig. 5), uniform damping with spectral radiation where $\kappa_O = 10^{-2}$ s$^{-1}$ and $\kappa_{O_3} = (3 \text{ months})^{-1} \approx 10^{-7}$ s$^{-1}$ (blue), uniform damping with gray radiation where $\sigma_{O_2} = 10^{-25}$ cm$^2$ molec$^{-1}$ and $\sigma_{O_3} = 5 * 10^{-18}$ cm$^2$ molec$^{-1}$ (gray), and the analytical approximation thereof from Eq. 28 (gray dashed).

where we used the nondimensional parameters defined in Eqs. 20-22. The UV flux is continuous across the transition altitude, but ozone is not generally continuous across $z_t$. Note that the ozone number density at $z_t$ is consistent between the photolytic sink regime (first line of Eq. 28) and the explicit solution at $z_t$ (second line of Eq. 28).

Fig. 7 shows the explicit solution to the gray ozone layer as it is developed through various approximations to the damping rate profile and spectrally-resolved radiative transfer. Compared to the full spectral Chapman+2 model, the gray analytical solution is partially degraded by each of it assumptions (vertically-uniform damping, gray radiation, and a sharp transition at $z_t$). The details of the solution are strongly dependent on the chosen parameters, which were selected both for their plausibility and their post hoc agreement with the observed profile. Rather than the details of the fit, the advantage of the gray solution is that it is arguably the first simple spectral model of ozone that affords an explicit solution to an ozone profile that has an interior maximum. It provides analytical expressions for the sensitivities of key aspects of the ozone layer to perturbations, noted briefly in the Discussion and to be explored in forthcoming work.



## 6 Discussion

### 6.1 Implications for understanding the response to perturbations

Distinguishing among competing theories for the same phenomenon can be justified, in part, if those theories make different predictions for the response to perturbations. This is the case among the three paradigms for explaining the ozone maximum. We illustrate this by considering the ozone response to a spectrally uniform doubling of top-of-atmosphere UV flux (holding $\kappa_O$, $\kappa_{O_3}$, and temperature fixed to only consider the direct effects on photolysis). Fig. 8a shows the response of the Chapman+2 model to such a doubling of UV, which leads to an increase of ozone primarily at and below the ozone maximum, which shifts downwards.

Analytical solutions using simple spectral models in each of the three paradigms predict qualitatively different responses from each other (Figs. 8b-d). In the source-controlled paradigm, doubling UV increases $O_3$ at all altitudes (as seen by decreasing $\alpha_{O_3}$ in Eq. B1). In the (unrealistic) strong-damping limit in which $O_3$ absorbs less than $O_2$, a doubling of UV leads to a doubling of $O_3$ at all altitudes (as seen by the textbook scaling in Eq. 12). Under weaker damping that is closer to realistic (in Fig. 8b, $\alpha_{O_3} = 10^{-3}$ s$^{-1}$), the increase in $O_3$ is more modest, because the increased $O_3$ aloft damps the effective UV perturbation towards the surface. Thus, in the source-controlled paradigm, the ozone increases are top-heavy, shifting the maximum upwards.

In the source/sink competition paradigm in the O-damped limit using a two-band approximation for the radiative transfer, doubling UV does not change $O_3$ at all (Fig. 8c), because it rescales the photolysis of $O_2$ and $O_3$ by the same factor. The constancy of $O_3$ under a spectrally-uniform rescaling of UV is also true for the fully spectral Chapman Cycle.

In the regime transition paradigm (Fig. 8d), as solved in the gray analytical theory (Eq. 28), the $O_3$ response brings together elements from both of the other paradigms. Ozone is in a photolytic sink regime above the ozone maximum, within which a doubling of UV leads to no change in $O_3$ at any given altitude. However, this increased UV speeds up O-damping at every altitude, thereby deepening the photolytic sink regime by shifting the transition altitude downwards. Due to the $n_a^3$ scaling of $O_3$ in the photolytic sink regime, this deepening of the photolytic sink regime increases $O_3$ below the control ozone maximum, and shifts the maximum downwards to a new transition altitude. Below this maximum, in the non-photolytic sink regime, the increased UV increases $J_{O_2}$, leading to increases in $O_3$ that are top-heavy within the non-photolytic sink regime. Therefore, just as in the Chapman+2 model, the regime transition paradigm predicts the largest increases in $O_3$ at and below the control ozone maximum, leading to a downward shift in that maximum.

### 6.2 The Chapman Cycle gets the right altitude of peak $O_3$ for the wrong reason

The Chapman Cycle predicts an interior maximum of $O_3$ at 25 km, almost exactly matching that assimilated in MERRA-2 of 27 km (Figs. 2 and 3d). The success of the Chapman Cycle at predicting the altitude of peak $O_3$ has underpinned its reputation as the foundational model of ozone photochemistry. Yet, in the Chapman+2 model, the source/sink competition paradigm (which explains interior maxima in the Chapman Cycle) predicted peak $O_3$ at 17 km (Fig. 5, red dashed curve), far below the Chapman Cycle prediction. The Chapman Cycle predicts the right altitude of peak $O_3$ for the wrong reason. This error can be elicited





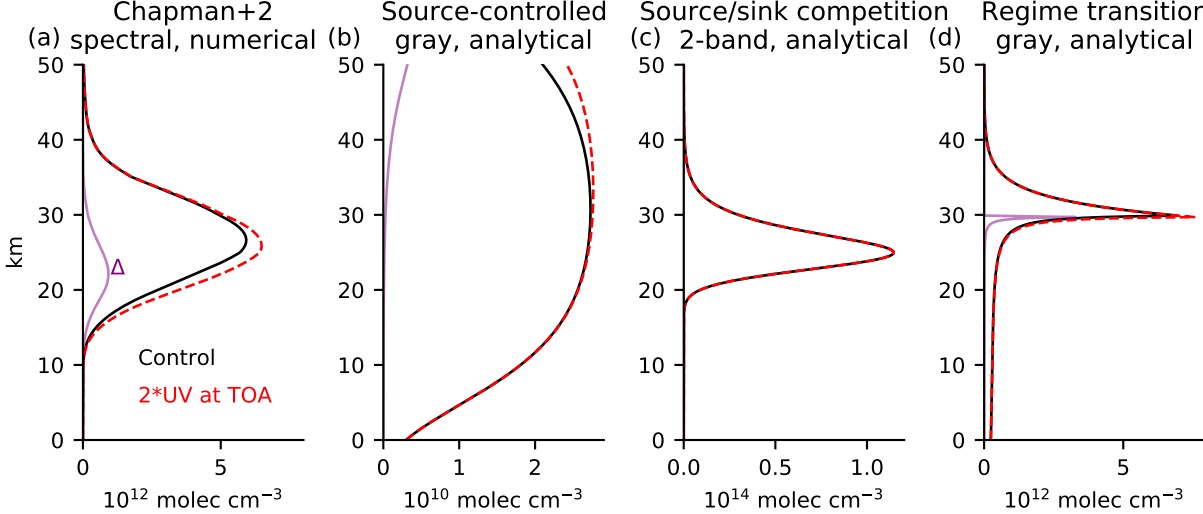

**Figure 8.** (a) Doubling the top-of-atmosphere UV flux ($I_\infty$) in the Chapman+2 model increases $O_3$ mainly below its control maximum, shifting the maximum downwards. Analytical solutions in each paradigm predict disparate responses to doubling UV: (b) in the source-controlled paradigm (Eq. B1), $O_3$ increases disproportionately aloft; (c) in the source/sink competition paradigm (Eq. B5), ozone remains constant; (d) in the regime transition paradigm (Eq. 28), $O_3$ increases below its control maximum, shifting that maximum downwards. Only the regime transition paradigm qualitatively reproduces the response in the Chapman+2 model.

by overwriting ozone in the Chapman Cycle with the reduced values from MERRA-2, and then calculating the UV fluxes and photolysis rates. With reduced $O_3$, the photolytic source/sink ratio is attenuated more slowly towards the surface, shifting the implied peak $O_3$ in the source/sink competition paradigm downwards to 18 km (Eq. 9). A similar effect manifests in the Chapman+2 model, with $O_3$ reduced by the generalized destruction. Thus, the Chapman Cycle's accurate altitude of peak $O_3$ results fortuitously from its overestimation of $O_3$, the amelioration of which actually shifts predicted peak $O_3$ to unrealistically low altitudes.

## 7   Conclusions

Of the ten textbooks analyzed here, seven explain the interior maximum within a source-controlled paradigm, within which ozone is argued to maximize where its source maximizes, at a sweet spot between abundant photons aloft and abundant $O_2$ below. Three textbooks explain the interior maximum within a source/sink competition paradigm adapted from the Chapman Cycle. In the source/sink competition paradigm, photolysis of ozone actively suppresses the concentration of ozone by producing atomic oxygen that can destroy ozone.

Both paradigms emerge as well-defined limits of the Chapman+2 model, a Chapman Cycle with damping of O and $O_3$ by generalized catalytic cycles and transport (Sec. 2). The source/sink competition paradigm corresponds to the O-damped limit,





and leads to a photolytic sink regime. The source-controlled paradigm corresponds to the $O_3$-damped limit, which leads to a non-photolytic sink regime. The tropical stratosphere was found to be in a photolytic sink regime above 26 km and in a non-photolytic sink regime below 26 km (Fig. 4).

That both paradigms are capable of explaining an interior maximum of ozone justifies why they have coexisted for so long. But, the Chapman+2 model reveals that neither paradigm can explain the observed tropical ozone maximum (Fig. 5). Instead, the observed ozone maximum arises due to a transition from a photolytic sink regime aloft to a non-photolytic sink regime below: a regime transition paradigm (Sec. 3, Fig. 6). This mechanism can be reproduced under gray radiation, leading to an explicit, piecewise-analytical solution for the ozone profile (Sec. 5). The regime transition paradigm shores up understanding

of the interior maximum of ozone in the tropical stratosphere and elucidates the response to an illustrative UV perturbation (Discussion). Forthcoming work will further investigate the response to perturbations.

## 8    Acknowledgments

The authors thank Daniel Cariolle for sharing the latest version (v2.9) of his linear ozone model. A.M. acknowledges constructive discussions with Benjamin Schaffer, Nadir Jeevanjee, Nathaniel Tarshish, and at the Princeton Center for Theoretical

Science workshop *From Spectroscopy to Climate*. This work was supported by the National Science Foundation under Award No. 2120717 and OAC-2004572, and by Schmidt Futures, a philanthropic initiative founded by Eric and Wendy Schmidt, as part of the Virtual Earth System Research Institute (VESRI).

## Appendix A:   Numerical details for solving the Chapman+2 model

We implement a numerical solution to the Chapman Cycle by solving Eq. 5 iteratively from the top of the atmosphere down-

wards. At any given level, we first solve for the UV flux reaching that level, which constrains the photolysis rates $J_{O_2}$ and $J_{O_3}$. These photolysis rates are then used to solve for $O_3$ (Eq. 5), which (along with $O_2$ constrains the UV flux reaching the level below. We consider the case of overhead sun. We consider generalized damping by prescribed parameters $\kappa_O$ and $\kappa_{O_3}$, but except as possibly accounted for by those damping rates, we do not explicitly account for advection, tropospheric chemistry, scattering, clouds, or surface reflection.

The vertical dimension is discretized into vertical levels ($\Delta z = 100$ meters) ranging from the surface to 100 km. The idealized shortwave radiative transfer and photolysis rates are solved on a wavelength grid with 621 discretized wavelengths ranging from 180 nm to 800 nm, extending into the Chappuis bands of weak absorption. Simulated absorption in the weakly-absorbing Chappuis bands ($\lambda > 400$ nm) is approximately $3*10^{-4}$ molec cm$^{-3}$ s$^{-1}$, consistent with that reported by Nicolet (1980). Spectrally-resolved parameters are linearly interpolated to the wavelength grid. Top-of-atmosphere UV flux is calculated from

the Solar Spectral Irradiance Climate Data Record (Coddington et al., 2015), averaged from 01-01-2020 to 02-04-2021 (Fig. 3a). $O_2$ absorption coefficients ($\sigma_{O_2}$) are taken from Ackerman (1971) and $O_3$ absorption coefficients ($\sigma_{O_3}$) from Sander et al.



(2010) (Fig. 3b). The isothermal atmosphere has a default temperature of 240 K and scale height of 7 km. Temperature-dependent parameters for reaction rates are taken from Brasseur and Solomon (2005).

## Appendix B: Simple spectral models for the Chapman Cycle

The interior maximum of ozone in the Chapman Cycle is of theoretical and historical significance (Chapman, 1930), yet clarity can still be gained as to how exactly this interior maximum comes about. The Chapman Cycle is in a photolytic sink regime, so its interior maximum is explained by the source/sink competition paradigm. We clarify the role of structure in the absorption coefficients in leading to this interior maximum by using two highly-idealized simple spectral models (SSMs) (terminology after Jeevanjee and Fueglistaler, 2020), for which we replace the $O_2$ and $O_3$ absorption spectra with simple analytic functions.

Once these analytic functions are embedded in the broader photochemical dynamics, we elucidate how the interior maximum of the ozone layer emerges from spectral absorption features.

### B1    No interior maximum under gray radiation

The Chapman Cycle can be solved explicitly in the limit of gray radiative transfer, just as in the case of the O-damped limit (Section 5.1), which also occupies a photolytic sink regime. In the gray limit, the photolytic source divided by the photolytic

sink reduces to the ratio of absorption coefficients, yielding an explicit expression for the gray Chapman Cycle ozone profile:

$$[O_3]_{\text{gray,Chapman}} = (\frac{\sigma^*_{O_2} k_2}{\sigma^*_{O_3} k_4})^{1/2} C_{O_2} n_a^{3/2} \tag{B1}$$

This explicit ozone profile can be integrated to yield a column ozone:

$$\chi_{O_3}(z) = \frac{2H}{3}[O_3]_{\text{gray,Chapman}} \tag{B2}$$

This expression for column ozone can be substituted into explicit expressions for the photolysis rates ($J_{O_2}$ and $J_{O_3}$). The

resulting gray Chapman Cycle solutions are shown in Fig. A1 (top row).

Because the production rate of ozone still maximizes as usual in the interior of the atmosphere but the concentration maximizes at the surface, the photolytic sink regime does not obey the source-controlled paradigm. The production rate of ozone ($J_{O_2}[O_2]$) maximizes at $\tau_{O_3} = 2/3$ even as $O_3$ maximizes at the surface. This reiterates that in the photolytic sink regime, ozone can maximize arbitrarily far below its source. Lifting the ozone maximum off the surface in the photolytic sink regime requires

spectral structure.

### B2    A two-band model for peak $O_3$ in the Chapman Cycle

Spectral structure can be incorporated with minimal complexity into our simple spectral model by adding an extra window of UV radiation, making this a two-band model. The key spectral structure is the *extension window* of ozone absorption at



higher wavelengths. The extension window results because $O_3$ can be photolyzed by lower-energy photons than $O_2$. $O_2$ can be photolyzed by ultraviolet with wavelengths up to 240 nm, whereas $O_3$ can be photolyzed by wavelengths beyond 240 nm into the visible, reflecting the weaker bonds of $O_3$ compared to $O_2$. Thus, below 240 nm there is absorption by both $O_2$ and $O_3$ in an *overlap window*, whereas beyond 240 nm there is only absorption by $O_3$ in the extension window.

We represent the extension window by extending $O_3$ absorption to longer wavelengths where it no longer overlaps with $O_2$ (Fig. A1d). Here, we assume that $O_3$ has the same absorption coefficient in the overlap and extension window, and that these two windows have equal width in wavelength. This additional absorption increases the photolysis rate of $O_3$:

$$J_{O_2} = \mu \sigma^*_{O_2} I_\infty \exp(-\sigma^*_{O_2} \chi_{O_2} - \sigma^*_{O_3} \chi_{O_3}) \tag{B3}$$

$$J_{O_3} = \mu \sigma^*_{O_3} I_\infty \exp(-\sigma^*_{O_2} \chi_{O_2} - \sigma^*_{O_3} \chi_{O_3}) + \mu \sigma^*_{O_3} I_\infty \exp(-\sigma^*_{O_3} \chi_{O_3}) \tag{B4}$$

The second term on the right-hand side of Eq. B4 is the additional photolysis in the extension window. Although $J_{O_2}$ has the same functional form as in the gray case, note that it will not take the same values because the $\chi_{O_3}$ refers to the overhead column ozone consistent with this particular photochemical solution. Plugging $J_{O_2}$ and $J_{O_3}$ into Eq. 9 again leads to cancellation of the implicit terms due to ozone attenuation and an explicit solution for ozone:

$$[O_3]_{\text{Extension}}(z) = \left( \frac{\sigma^*_{O_2} k_2}{\sigma^*_{O_3}(1 + \exp(\sigma^*_{O_2} \chi_{O_2}(z))) k_4} \right)^{1/2} C_{O_2} n_a(z)^{3/2} \tag{B5}$$

This is an explicit expression for an ozone profile with an interior maximum in the Chapman Cycle using the two-band SSM. The solution depends on overhead column $O_2$ (assumed invariant). Whereas the Gray SSM had constant $J_{O_2}/J_{O_3}$ with height, the Extension SSM has $J_{O_2}/J_{O_3}$ decreasing towards the surface. In the limit where $\exp(\sigma^*_{O_2} \chi_{O_2}) \gg 1$, the maximum number density of ozone occurs at $\tau_{O_2} = 3$. For the parameters in Fig. A1f, this maximum occurs at 17 km. The altitude of peak $O_3$ depends only on $O_2$ optical depth because, with constant $\sigma^*_{O_3}$, absorption by $O_2$ is what causes the photolytic source to attenuate faster than the photolytic sink.

Conceptually, in the photolytic sink regime, ozone maximizes in the interior of the atmosphere due to competition between the exponentially-increasing air density towards the surface and the declining ratio of the photolytic source to the photolytic sink ($J_{O_2}/J_{O_3}$). The Extension SSM reveals that the photolysis rate of $O_2$ is attenuated faster than the photolysis rate of $O_3$ due to the joint structure of the $O_2$ and $O_3$ absorption coefficients, which have a region of overlapping absorption that both produces and destroys ozone and a region of extended ozone absorption that only destroys ozone. Once the overlap window saturates with $O_2$, its contribution to both the ozone source and sink begins to decline rapidly. Because the overlap window accounts for all of the source but only part of the sink, the sink being buttressed by contributions from the extension window, the source decreases relative to the sink.

The results from the Extension SSM suggest that the interior maximum in the photolytic sink regime is explained by the source/sink competition paradigm. Our analytical expression provides rigorous support for previous explanations within the



source/sink competition paradigm. For example, Dutsch (1968) wrote (with adapted notation), "The formation of a layer
535  of maximum ozone content arises from the fact that below about 35 km the dissociation rate of molecular oxygen ($J_{O_2}$)
drops off much more rapidly than that of ozone ($J_{O_3}$), mainly because of the overlap of ozone and oxygen absorption around
210 nm." McElroy (2002) wrote that the concentration of $O_3$ "is small at low altitudes, reflecting the *comparative absence*
[emphasis added] of radiation with wavelengths sufficiently short to effect dissociation of $O_2$." "Comparative" refers to the
difference between the ozone production and destruction. The evidence from the Gray and Extension SSMs places these
previous arguments on a firmer foundation for two reasons: (1) it interventionally isolates the role of the overlap and extension
windows in leading to ozone structure, and (2) it provides an explicit solution for ozone where these previous arguments were
based on explaining ozone in diagnostic terms from the inferred photolysis rates, which are implicit functions of ozone.

**Appendix B: An explicit gray solution in the non-photolytic sink regime**

In Section 5.3, we derived the ozone profile in a non-photolytic sink regime below some transition altitude $z_t$ at which $J_{O_2}(z_t)$
was known. Here, we derive an ozone profile for an atmosphere assumed to be everywhere in a non-photolytic sink regime.
Our derivation can be generalized from that in Section 5.3 by taking $z_t$ towards $\infty$ and substituting $J_{O_2}(z_t)$ as dictated by the
top-of-atmosphere UV flux, i.e., $J_{O_2}(\infty) = \sigma^*_{O_2} \mu I_\infty$. This yields the following expression for ozone:

$$[O_3](z) = \frac{\sigma^*_{O_2} C_{O_2} n_{a_0} \exp(-z/H)}{\sigma^*_{O_3} \left( (1 + \alpha_{O_3}) \exp(\tau_{O_2}(0) \exp(-z/H)) - 1 \right)} \tag{B1}$$

where the non-dimensional parameters $\alpha_{O_3}$ and $\tau_{O_2}(0)$ were defined by Eqs. 21 and 22. The values of $\alpha_{O_3}$ must be restricted
by the assumption that damping is strong enough to lead to a non-photolytic sink regime, which rules out values of $\alpha_{O_3}$ below
a certain threshold that can be *post hoc* verified for a given solution.

By differentiating Eq. B1, the ozone maximum can be found to be located at the following optical depth with respect to $O_2$:

$$\tau_{O_2, \text{max } O_3} = W\left( \frac{-1}{(1 + \alpha_{O_3})e} \right) + 1 \tag{B2}$$

Eq. B2 reveals that when damping is very strong, in the limit of $\alpha_{O_3}$ going to $\infty$, the interior maximum of ozone is at
$\tau_{O_2} = 1$, i.e., at the sweet spot calculated from $O_2$ absorption. This limit corresponds to the limit of vanishing ozone, in which
$O_2$ is the dominant absorber of UV, recovering the textbook problem from Jacob (1999) that neglects $O_3$ (Eq. 12). However, as
damping weakens to the point that $O_3$ increases enough to become the dominant absorber, while still ensuring that the damping
is strong enough to be in the non-photolytic sink regime, absorption by ozone suppresses the production rate at lower altitudes
and shifts the interior maximum in ozone production towards higher altitudes. Consequently, the interior maximum of ozone
also shifts upwards. Absorption by ozone shifts the interior maximum of ozone to higher altitudes than when such absorption
is neglected.

Fig. C1 shows how the theoretical scaling compares with numerical solutions to the monochromatic Chapman Cycle with
$O_3$ damping. The theoretical scaling correctly captures that, for strong damping, the ozone maximum approaches $\tau_{O_2} = 1$,




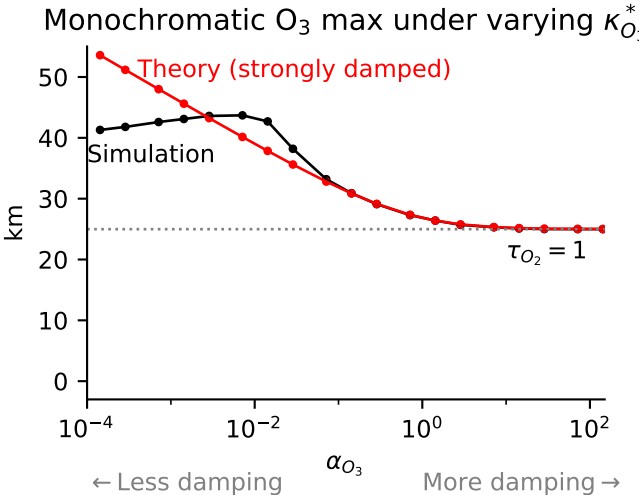

**Figure A1.** Chapman Cycle photochemical equilibrium in simple spectral models of the ozone layer (first two rows) and the full spectral model (bottom row). (Left column) Absorption coefficients in each model (solid) compared to in reality (transparent). (Middle column) $O_2$ and $O_3$ photolysis rates. (Right column) Ozone number density.

which is the gray $O_2$-only limit based on Eq. 12. As damping is reduced, the theoretical scaling correctly captures that the interior maximum shifts upwards as absorption by ozone aloft reduces the ozone production rate at lower altitudes. However, further reductions in damping lead to the violation of the underlying assumptions of the theoretical scaling, namely that ozone is everywhere in the non-photolytic sink regime. Instead, the Chapman Cycle sink of ozone can dominate in the upper atmosphere, leading to a photolytic sink regime aloft that is not accounted for within this theory. Thus, the theory performs well in its range of applicability, but does not explain the observed ozone maximum on Earth.

**Figure C1.** Altitude of maximum ozone in a monochromatic ozone solution with absorption by $O_2$ and $O_3$ and varying damping $\kappa_{O_3}^*$ that modulates the nondimensional $O_3$-damping parameter $\alpha_{O_3}$. Comparison of numerical simulations (magenta) with analytical theory (Eq. B2). The theory reproduces the large-damping limit of $\tau_{O_2} = 1$. As damping is weakened, ozone absorption aloft shifts ozone maximum upwards. As damping is further weakened, the theoretical assumption of a non-photolytic sink regime breaks down, degrading its applicability to the simulation.

*Code and data availability.* The Chapman Cycle Photochemical Equilibrium Solver described in Section 2 is published at doi:10.5281/zenodo.10515739.

*Author contributions.* Authors' contributions: AM and EPG acquired funding; AM, EPG, and SF conceptualized research; AM performed formal analysis; AM wrote original draft; EPG and SF reviewed and edited paper.

*Competing interests.* The authors declare that they have no conflict of interest.

*Acknowledgements.* The authors thank Daniel Cariolle for sharing the latest version (v2.9) of his linear ozone model. A.M. acknowledges constructive discussions with Benjamin Schaffer, Nadir Jeevanjee, Nathaniel Tarshish, and at the Princeton Center for Theoretical Science workshop *From Spectroscopy to Climate*. This work was supported by the National Science Foundation under Award No. 2120717 and OAC-2004572, and by Schmidt Sciences, a philanthropic initiative founded by Eric and Wendy Schmidt, as part of the Virtual Earth System Research Institute (VESRI).




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
