# Peer review of "Protection without poison: Why tropical ozone maximizes in the interior of the atmosphere"

_EGUsphere, 2024_

## Referee Comment (RC1)

**Review of "Protection without poison: Why tropical ozone maximizes in the interior of the atmosphere" by A. Match et al.**

This study seeks to construct a minimal theory that would account for the ozone peak at ~26 km and provide a simple conceptual explanation of the peak location. This is accomplished by introducing a "Chapman+2 model" in which the Chapman reactions are augmented by two additional generalized "reactions" that account for ozone destruction by catalytic cycles and by transport, and subsequent analysis of the limit cases. The paper shows that these limit cases correspond to two conceptual explanations of the tropical ozone maximum, neither of which produces the maximum at the correct altitude. Instead, the authors show that the true maximum is located at the transition altitude between the two regimes in agreement with data.

Furthermore, the authors develop an explicit solution to ozone photochemistry under simplified assumptions (gray atmosphere) that produces an internal ozone peak at the correct altitude and in agreement with the proposed "regime transition paradigm".

I think there is a pedagogical value in being able to provide an explicit formula for the tropical ozone profile, of the form $[O_3] = \ldots$, even if that formula (Eq. 28) is not exactly simple and probably(?) has to be evaluated numerically because it includes the Lambert W function. Still, it tells a consistent story of the proposed regime transition paradigm governing the location of the ozone peak. The paper is very clearly written. The math, while tedious in places, is explained in sufficient detail for the readers to follow the argument. I think this is a very nice contribution to stratospheric ozone science and clearly well within the scope of ACP.

I have only minor comments and suggestions, some more pedantic than the others.

Good work!
Kris Wargan

**Minor and technical comments**

L20-21. Do you need this distinction? There are other radiatively active (in the sense of having absorption spectra) gases that maximize "in the interior", e.g. $HNO_3$ (at midlatitudes in this case). Also, I wouldn't consider $CH_4$ well mixed. Other long-lived species, such as $N_2O$, CFCs etc. are neither well mixed nor confined to near the surface.

L50. MERRA-2 stands for "Modern Era Retrospective Analysis for Research and Applications". Please, cite at least the MERRA-2 core paper, Gelaro et al., 2017, and maybe also the MERRA-2 ozone description and evaluation paper, Wargan et al., 2017. Is this a zonal average? Climatology? Over what latitudes and time? MERRA-2 ozone is at its best after ~2004 when MLS is assimilated.

L87. Typo: "which we then augment it with". Drop the "it"

L102. It's not entirely accurate to say that it's "photolysis" that does the attenuation. Doesn't scattering also play a role?

L120-127. Why is it OK to neglect the effects of mixing / leaky tropical pipe?

L130 / R5 & R6. Why is it justified to treat vertical advection as an effective chemical loss? In particular, L133-134 imply a linear dependence of the generalized loss on the concentrations of O and $O_3$. But advection by $\overline{\omega *}$ is proportional to the vertical gradient of the mixing ratio (equivalently, the appropriate term in the continuity equation), not to concentration. So, why does it work? This is actually discussed in Section 3.5.2 of Brasseur and Solomon 2005, but I think it should be briefly explained here too, especially that this approximate treatment of transport maybe responsible for some remaining inaccuracies of the Chapman+2 model, as later explained in the discussion of Fig. 3.

Eqs. 5 and 6. I hesitate to ask for more math;) However, it may be instructive to see the full set of continuity equations from which these are derived under a steady state assumption. It could be another short appendix. It's just a suggestion. A committed reader should be able to back out the full set of equations from 5 and 6.

L142. "implicit ... due to dependence on ozone aloft". So, not really quadratic in $[O_3]$ to leading order as stated earlier in the same sentence? Or is the dependence of the photolysis rates on integrated ozone sub-quadratic/weak? Is any of this relevant to the calculations?

LL156-157. See my comment to L130 above.

L186. I would imagine that the diurnal cycle at those altitudes is pretty much negligible compared to the other factors.

L197-199 and 210. Nice!

L 202. "Chapman dynamics"? Or chemistry?

Eq 11. Shouldn't there be a factor of $n_a$ in the numerator? Equation 5 has an $n_a^3$ in the prefactor and $n_a^2$ in the relevant term in the denominator.

L242. Typo: duplicate "regimes"

L247. As above. Is this a climatological mean tropical profile from MERRA-2? A profile for a specific day?

L250-251, Fig 5. I suggest cutting the figure off at the approximate altitude of the tropopause. Everything below that is irrelevant to this discussion.

L271 and below. Is "non-analytic" used in the mathematical sense of not having aTaylor expansion or in the loose sense (no closed formula)? In the latter case I think people say "non-analytical". I may be wrong.

L351, 354. Since Eq. 23 contains the Lamber W-function it may be questioned whether this qualifies as an analytical solution. I'm not sure if there's a closed formula for it even in this case

where the argument is real and positive. As far as I know W is evaluated numerically using rather terrifying-looking approximations (e.g. https://mathworld.wolfram.com/LambertW-Function.html).

L390. The profiles in Fig. 7 look continuous to me. What am I missing?

L543. I think it's supposed to be Appendix C.

**References**

Gelaro, R., McCarty, W., Suárez, M. J., Todling, R., Molod, A., Takacs, L., et al. (2017). The Modern-Era Retrospective analysis for research and applications, version 2 (MERRA-2). Journal of Climate, 30(14), 5419–5454. https://doi.org/10.1175/JCLI-D-16-0758.1

Wargan, K., Labow, G., Frith, S., Pawson, S., Livesey, N., & Partyka, G. (2017). Evaluation of the ozone fields in NASA's MERRA-2 reanalysis. Journal of Climate, 30(8), 2961–2988. https://doi.org/10.1175/JCLI-D-16-0699.1

---

## Referee Comment (RC2)

*Review of*

**"**Protection without poison: Why tropical ozone maximizes in the interior of the atmosphere** "**

*by A. Match et al.*

**General**

I think it is good that this paper addresses issues of catalytic cycles of ozone loss and ozone production in the tropical stratosphere. The study goes back to textbooks and examines what can be learned from the 'classic' explanations and in how far they should perhaps be modified.

However, the paper needs to have a clearer message. I do not think that the fact that tropical ozone has a peak (in concentration) at about 26 km is the new finding that this papers wants to report. And also not the concentration of ozone at this altitude. As I read the paper, what is new here is the advance in understanding of several issues that is gained through the different simplified models. I do not think this aspect is coming across very well.

The analysis in the paper also based on certain assumptions, which are important for the presented derivations. I suggest mentioning these assumptions clearer and upfront (rather than in the course of particular derivations). There are also questions about the "real world" ozone representation (i.e., MERRA-2, see also below).

Overall I suggest improvements to the manuscript. The message of the paper should come across more clearly.

**Comments**

**Applicability of the analysis**

Clearly, the maximum (in concentration) of ozone is not the same everywhere, in particular it depends on latitude. It is not clear which region the paper is addressing, looking at section 2 it seems the entire atmosphere (but the the maximum is more than one value); however the title says 'tropical'. Further, the maximum of

ozone concentration at a particular altitude (even for a given latitude) can only be considered a fixed number for some kind of climatology (i.e., for some averaging; see also below). This aspect should be addressed in the paper, e.g., there should be more information in the caption of Fig. 2 (which latitude range, which period? etc., see also comments on Fig. 3). Transport is represented rather crudely (l. 120-127) and the analysis is restricted to the tropical lower stratosphere. So is the tropical (lower) stratosphere the regime to which this analysis should be applied? I think the answer is 'yes', but this should be clearly stated throughout the paper.

Another aspect is tropospheric ozone. At some instances in the paper it is clearly stated that this analysis is not about tropospheric ozone chemistry (l. 468). This is correct and I agree. But this is not clearly and upfront stated in the paper. And some of the figures (e.g. 2, 4, 5) extend to the ground, leaving the impression that the discussion in the paper is relevant also for tropospheric altitudes (for which is it not). Looking at Figs. like Fig. 5, it is clear that the real atmosphere (here MERRA-2, see also below) does not look like anything in the 'non-photolytic sink regime' below the tropopause. Moreover, given the fact that the tropical tropopause is at about 18 km (Hoffmann and Spang, 2022); most of the the 'non-photolytic sink regime' in Fig. 5 is irrelevant.

**Damping**

Throughout the paper, the concept of "damping" is used. It is mainly applied to ozone ($O_3$) or to O (where $O_x = O + O_3$). To me "damping" is the energy loss of an oscillating system through dissipation. Here this concept seems to be used as a synonym for change of $O_3$ or O at a particular location through chemical (catalytic) loss and advection of $O_3$ (e.g., advection of low ozone from below in the tropical lower stratosphere).

The concepts of chemical change and advection of stratospheric ozone are well known (see e.g. the textbooks cited in this manuscript), so I do not really see the advantage of redefining these concepts in terms of "damping". At least, I suggest that the wording used here is related to the well established concepts of catalytic loss and advection of stratospheric ozone. Furthermore, these concepts apply to ozone in photochemical equilibrium, where $\partial O_3/\partial t = \partial O/\partial t = 0$ – correct?

**Twenty six kilometres**

Throughout this paper (starting with the abstract, ls. 2, 10) the altitude of the peak of the ozone layer (in terms of concentration) is a key point here. And the transition between two photochemical regimes. Twenty six kilometres is the number that is reported. However, how certain is this number; no error estimate is given, what is the uncertainty for the 26 km? Also, is there a variability of the 26 km with other atmospheric parameters (e.g. QBO, Nivano et al., 2003; Diallo et al., 2018).

Perhaps more importantly, any successful simulation of the tropical ozone layer should not only reproduce the location of the peak in altitude but also the vertical distribution in ozone concentration with altitude (including the ozone concentrations at the peak); I think this aspect could be treated more thoroughly in this paper.

**Assumptions**

There are a number of assumptions on ozone made in this study that are mentioned; however I suggest to state these assumptions more clearly ad upfront. For example, photochemical equilibrium of ozone is assumed ($\partial[O_3]/\partial t = 0$; l. 138), which is a strong assumption. This assumption is not valid for a large part to the stratosphere. (Which seems to be the reason why this analysis is restricted to the tropics). Further there are assumptions like an isothermal atmosphere (also not realistic) and other assumptions (l. 293) that should be clear.

**Catalytic cycles destroying ozone**

The Chapman (1930) model is known to be incorrect insofar as it neglects the most relevant catalytic ozone loss cycles (e.g., Portmann et al., 2012). Thus to investigate tropical ozone (which seems to be the target here) one needs to look at the tropical profiles for the relevant species driving ozone loss in the tropics. I do not think this is the case here and I recommend changing it. Global estimates are not helpful here, the atmosphere is very different in the tropics and in the mid-latitudes. Already decades ago, researchers (including the editor of the present manuscript) have invested substantial effort into deriving ozone loss cycles in the tropics (e.g., Crutzen et al., 1995).

**Formation and destruction of ozone**

In the introduction, there is a discussion of the ozone sources and sinks in the atmosphere. While I like the idea to go back to the textbooks, it should be clear that much more is known today than what is discussed in Fig. 1. Some modern textbooks are cited, but another example is Portmann et al. (2012). Regarding the history of the debate on the relevant sinks of ozone in the atmosphere I also recommend the book by Brasseur (2019).

**Odd oxygen**

Commonly, odd oxygen $O_x$ is defined as $O_x = O + O_3$. This is also done here. This is a well established concept (see e.g. the textbooks cited in this manuscript). However, $O_x$ is used here first (l. 132) before being defined (l. 136). But more importantly, the concept needs to be introduced briefly before being used (even if the reviewer has learned about $O_x$ before). Further, below (l. 163) $O_y$ is mentioned – but without knowing what this is, any discussion below on $O_y$ is not very useful. Define in this paper what $O_y$ is.

**Ozone in the real world**

As a reference for ozone in the real atmosphere, MERRA-2 (Gelaro et al., 2017) is used. Always one particular profile is presented , but it is not clear what the profile shows. In the caption of Fig. 3, one reads that this is a tropical ozone profile (30°S to 30°N), but this is not the most conservative estimate of the tropics; perhaps 20°-20° would be more appropriate? Has this sensitivity been explored? The profile is for 2018. I could not find another place in the manuscript where the profile is explained (which should be changed). I assume that it is a zonal mean profile and that the profile is annually averaged (of course, I am not sure). But this should be clearly stated in the manuscript. Another question is, which vertical resolution was used for MERRA-2 ozone.

Further, how was 2018 chosen? As pointed out in the manuscript tropical ozone depends on the solar UV-flux, which is changing with the 11-year solar cycle (albeit not by a factor of two). Tropical ozone might also be influenced by the QBO (see above); could this point be of interest here? If only one single year is considered, why was a reanalysis chosen rather than direct measurements (e.g. MLS, Waters et al., 2006; Han et al., 2019). A climatology based on (ozone)

observations could be an alternative (e.g., SWOOSH, Davis et al., 2016).

**Technical issues**

The paper could be easier to read. There are a few issues where a clearer language would help. The paper talks about a "gray ozone layer" – to me this is jargon. This term (gray ozone) is not familiar to the readers of ACP; it is rather a shorthand for assumptions about the lack of a spectral dependence of $\sigma_{O_3}$ and $\sigma_{O_2}$.

In Eq. 7 (l. 160) $a_5$ (and similar coefficients) are used. I cannot see that these coefficients have been defined before (did I overlook anything?). It would be good to explain here what the atmospheric meaning of $a_5$ (etc.) is.

Also, $O_y$, is used in the paper, but this is not an established notation (see also the comments on odd oxygen above). I also suggest a better introduction to the concept of 'damping' to make the paper more easily accessible (again see discussion above). Further, there is 'appendix B' two times in the paper – this is confusing.

There are recommendations by ACP; the abstract is likely too long. Also, it would be good if the coefficients for the Cariolle 2.9 scheme used here were available – in this way the results of this paper could be reproduced and the coefficients be used for other purposes.

**Minor issues**

- l. 17: This is a matter of opinion, but I suggest not attributing the discovery of the ozone layer to Hartley alone; see the discussion by Brasseur (2019).

- l. 35: I suggest also to have a look at the classic textbook by Dobson (1963). While the textbooks want to give a simple message to the reader, it is clear that production of ozone alone, without a loss mechanism for ozone would simply convert $O_2$ to ozone. This is why Dobson (1963), on page 105 of his book, calls the chapter "FORMATION AND DESTRUCTION OF OZONE".

- l. 50: The quoted MERRA-2 ozone profile is for the tropics (information only in the caption). But how are the tropics defined here? For which period is the ozone profile valid? Likely the MERRA-2 ozone is the same as the

one in Fig. 3 – correct? Does MERRA-2 assimilates ozone observations? Perhaps add a reference for MERRA-2 (see above).

- l. 64: Is is clear here what passive and active sinks are?

- l. 83: 'gray radiative transfer' is not clear to me here (see also above).

- l. 83: 'endogenously' sounds like a medical term to me – is it really helpful here?

- eq. 3: larger brackets for the exponential function (also elsewhere)

- l. 122: 'damping ozone' could be better explained

- l. 128: is this 'augmentation' meant to be globally or tropical?

- l. 130: I think the coefficients $z_O$ and $z_{O_3}$ are important for this paper, but they are not reported (did I miss anything?)

- Fig. 3: I would not call MOBIDIC a chemistry climate model; I think nowadays something else is understood by the term chemistry climate model.

- l. 157: how is this vertical scale determined?

- l. 167: I do not understand, why 'globally averaged' profiles of chemical constituents are used. The paper is on tropical ozone.

- l. 176 'eff' should not be in italics

- l. 182: more need to be explained here that 'average' – see also above.

- l. 187: I do not think that it is necessary to use globally averaged profiles here. (Also the profile is probably not 'catalytic'.)

- l. 197: Does the Chapman cycle sink ever dominate in the atmosphere?

- l. 216: adding the reactions that lead to the 'domination' would be helpful here.

- Fig. 4: It should be clear that this figure is for the tropics. Second, the figure extends to the ground, but the tropospheric chemistry prevailing below $\approx$ 18 km is not discussed here.

- l. 241: It is nor clear here where the 26 km value comes from.

- l. 270: solutions are only for equations, not for a 'layer'. (Also l. 294).

- l. 298: is it clear here *why* the transition altitude is the altitude of the ozone maximum?

- l. 314: the surface is not a region where these theories should be applied.

- l. 430, 431: I am confused here: the red line in Fig. 5 is discussed (source/sink paradigm) – is does not provide a good estimate for the ozone maximum. However the Chapman+2 model (magenta line) does. So why are we concerned about the red line if the Chapman+2 model seems appropriate?

- l. 436: 'generalized destruction' is not clear.

- l. 473: change $*$ to \cdot

- l. 476: why are the absorption coefficients not taken from the most recent kinetic recommendation (Burkholder et al., 2020)?

- l. 477: Why is it not possible to approximately take the atmospheric temperature profile into account when calculating temperature dependent kinetic parameters?

- Fig. C1: I cannot see the magenta line mentioned in the caption in this figure.

- l. 614: 'The atmospheric environment' is listed here twice.

**References**

Brasseur, G. P.: The Ozone layer: From Discovery to Recovery, American Meteorological Society, 2019.

Burkholder, J. B., Sander, S. P., Abbatt, J. P. D., Barker, J. R., Cappa, C., Crounse, J. D., Dibble, T. S., Huie, R. E., Kolb, C. E., Kurylo, M. J., Orkin, V. L., Percical, C. J., Wilmouth, D. M., and Wine, P. H.: Chemical kinetics and photochemical data for use in atmospheric studies, Evaluation Number 19, JPL Publication 19-5, URL `http://jpldataeval.jpl.nasa.gov`, 2020.

Chapman, S.: A theory of upper atmospheric ozone, Mem. Roy. Soc., 3, 103–109, 1930.

Crutzen, P. J., Grooß, J.-U., Brühl, C., Müller, R., and Russell III, J. M.: A Ree-valuation of the ozone budget with HALOE UARS data: No evidence for the ozone deficit, Science, 268, 705–708, 1995.

Davis, S. M., Rosenlof, K. H., Hassler, B., Hurst, D. F., Read, W. G., Vömel, H., Selkirk, H., Fujiwara, M., and Damadeo, R.: The Stratospheric Water and Ozone Satellite Homogenized (SWOOSH) database: a long-term database for climate studies, Earth System Science Data, 8, 461–490, https://doi.org/10.5194/essd-8-461-2016, 2016.

Diallo, M., Riese, M., Birner, T., Konopka, P., Müller, R., Hegglin, M. I., Santee, M. L., Baldwin, M., Legras, B., and Ploeger, F.: Response of stratospheric water vapor and ozone to the unusual timing of El Niño and the QBO disruption in 2015–2016, Atmos. Chem. Phys., 18, 13 055–13 073, https://doi.org/10.5194/acp-18-13055-2018, 2018.

Dobson, G. M. B.: Exploring the atmosphere, Oxford University Press, 1963.

Gelaro, R., McCarty, W., Suárez, M. J., Todling, R., Molod, A., Takacs, L., Randles, C. A., Darmenov, A., Bosilovich, M. G., Reichle, R., Wargan, K., Coy, L., Cullather, R., Draper, C., Akella, S., Buchard, V., Conaty, A., da Silva, A. M., Gu, W., Kim, G.-K., Koster, R., Lucchesi, R., Merkova, D., Nielsen, J. E., Partyka, G., Pawson, S., Putman, W., Rienecker, M., Schubert, S. D., Sienkiewicz, M., and Zhao, B.: The Modern-Era Retrospective Analysis for Research and Applications, Version 2 (MERRA-2), jci, 30, 5419–5454, 2017.

Han, Y., Tian, W., Chipperfield, M. P., Zhang, J., Wang, F., Sang, W., Luo, J., Feng, W., Chrysanthou, A., and Tian, H.: Attribution of the Hemispheric Asymmetries in Trends of Stratospheric Trace Gases Inferred From Microwave Limb Sounder (MLS) Measurements, J. Geophys. Res., 124, 6283–6293, https://doi.org/https://doi.org/10.1029/2018JD029723, 2019.

Hoffmann, L. and Spang, R.: An assessment of tropopause characteristics of the ERA5 and ERA-Interim meteorological reanalyses, Atmos. Chem. Phys., 22, 4019–4046, https://doi.org/10.5194/acp-22-4019-2022, 2022.

Nivano, M., Yamazaki, K., and Shiotani, M.: Seasonal and QBO variations of as-cent rate in the tropical lower stratosphere as inferred from UARS HALOE trace

gas data, J. Geophys. Res., 108, 4794, https://doi.org/10.1029/2003JD003871, 2003.

Portmann, R. W., Daniel, J. S., and Ravishankara, A. R.: Stratospheric ozone depletion due to nitrous oxide: influences of other gases, Phil. Trans. R. Soc. B, 367, 1256–1264, https://doi.org/10.1098/rstb.2011.0377, 2012.

Waters, J. W., Froidevaux, L., Harwood, R. S., Jarnot, R. F., Pickett, H. M., Read, W. G., Siegel, P. H., Cofield, R. E., Filipiak, M. J., Flower, D. A., Holden, J. R., Lau, G. K., Livesey, N. J., Manney, G. L., Pumphrey, H. C., Santee, M. L., Wu, D. L., Cuddy, D. T., Lay, R. R., Loo, M. S., Perun, V. S., Schwartz, M. J., Stek, P. C., Thurstans, R. P., Boyles, M. A., Chandra, S., Chavez, M. C., Chen, G.-S., Chudasama, B. V., Dodge, R., Fuller, R. A., Girard, M. A., Jiang, J. H., Jiang, Y., Knosp, B. W., LaBelle, R. C., Lam, J. C., Lee, K. A., Miller, D., Oswald, J. E., Patel, N. C., Pukala, D. M., Quintero, O., Scaff, D. M., Snyder, W. V., Tope, M. C., Wagner, P. A., and Walch, M. J.: The Earth Observing System Microwave Limb Sounder (EOS MLS) on the Aura satellite, IEEE Trans. Geosci. Remote Sens., 44, 1106–1121, 2006.

---

## Author Comment (AC1)

**Response to Reviewers: "Protection without poison: why tropical ozone maximizes in the interior of the atmosphere" (egusphere-2024-1552)**

Aaron Match, Edwin P. Gerber, and Stephan Fueglistaler

October 21, 2024

We appreciate the thoughtful reviews of our first submitted version of "Protection without poison: why tropical ozone maximizes in the interior of the atmosphere". The reviewers' supportive as well as critically constructive comments have encouraged us to improve the manuscript. We believe that through extensive revisions of the manuscript for clarity and extensive sensitivity tests of our methodology included in this review, we have addressed the comments of both reviewers, resulting in a stronger manuscript.

Several general revisions to the manuscript are that, in the interest of adding more clarity for the reader, we have created a new section called "Why there is a regime transition", which comes with a new Figure 7. Also, we have sharpened Figure 2 and the Introduction, reordered Section 2 introducing the Chapman+2 model, noted new considerations about the partitioning between $NO_x$ and $HO_x$ sinks near the stratopause, and streamlined the Section deriving the analytical ozone profile.

We again thank the reviewers. Throughout this Response to Reviewers, reviewer comments will be in black and our author comments will be in blue.

**1 First Review**

This study seeks to construct a minimal theory that would account for the ozone peak at ∼26 km and provide a simple conceptual explanation of the peak location. This is accomplished by introducing a "Chapman+2 model" in which the Chapman reactions are augmented by two additional generalized "reactions" that account for ozone destruction by catalytic cycles and by transport, and subsequent analysis of the limit cases. The paper shows that these limit cases correspond to two conceptual explanations of the tropical ozone maximum, neither of which produces the maximum at the correct altitude. Instead, the authors show that the true maximum is located at the transition altitude between the two regimes in agreement with data.

Furthermore, the authors develop an explicit solution to ozone photochemistry under simplified assumptions (gray atmosphere) that produces an internal ozone peak at the correct altitude and in agreement with the proposed "regime transition paradigm".

I think there is a pedagogical value in being able to provide an explicit formula for the tropical ozone profile, of the form $[O_3] = ...$, even if that formula (Eq. 28) is not exactly simple and probably(?) has to be evaluated numerically because it includes the Lambert W function. Still, it tells a consistent story of the proposed regime transition paradigm governing the location of the ozone peak. The paper is very clearly written. The math, while tedious in places, is explained in sufficient detail for the readers to follow the argument. I think this is a very nice contribution to stratospheric ozone science and clearly well within the scope of ACP.

I have only minor comments and suggestions, some more pedantic than the others.

Good work! Kris Wargan

Kris, we agree with your summary of our contribution, and greatly appreciate your support as well as your minor comments and suggestions.

**1.1 Minor and technical comments**

L20-21. Do you need this distinction? There are other radiatively active (in the sense of having absorption spectra) gases that maximize "in the interior", e.g. $HNO_3$ (at midlatitudes in this case). Also, I wouldn't consider $CH_4$ well mixed. Other long-lived species, such as $N_2O$, CFCs etc. are neither well mixed nor confined to near the surface.

You have raised the issue that whether or not a constituent is thought of as well-mixed depends on the region of interest and the timescale of the processes of interest. Following this comment, we have reframed the introduction to omit this distinction.

L50. MERRA-2 stands for "Modern Era Retrospective Analysis for Research and Applications". Please, cite at least the MERRA-2 core paper, Gelaro et al., 2017, and maybe also the MERRA-2 ozone description and evaluation paper, Wargan et al., 2017. Is this a zonal average? Climatology? Over what latitudes and time? MERRA-2 ozone is at its best after ~2004 when MLS is assimilated.

Thanks for the clarification, and we would not have wanted these valuable products to get anything less than the recognition they deserve. Although MERRA-2 served our purposes of estimating the average location of the interior maximum of ozone to within approximately 1 km, in light of the concerns raised by both reviewers about the MERRA-2 profile, we have opted to use the homogenized observational dataset SWOOSH (Davis et al., 2016) averaged from 1984-2023 and from 30°S-30°N. SWOOSH has approximately 1 km vertical resolution near the interior maximum of $[O_3]$, and the interior maximum is located at the vertical level closest to 26 km during approximately 90% of the time in this monthly dataset.

L87. Typo: "which we then augment it with". Drop the "it" Corrected.

L102. It's not entirely accurate to say that it's "photolysis" that does the attenuation.

Doesn't scattering also play a role? This raises an important point. Among the many physical and photochemical simplifications of this work, and in our efforts to distill only those processes that are essential for the ozone maximum, our Chapman+2 model framework neglects scattering and instead uses an optical depth-based radiative transfer where all attenuation is assumed to be absorption. This approximation is consistent with standard pedagogical explanations for the Chapman cycle of the interior maximum of ozone, which do not emphasize the role of scattering in ozone layer structure, e.g., Jacob (1999). It might be interesting in future work to analyze any effects of scattering on the structure of the ozone layer.

L120-127. Why is it OK to neglect the effects of mixing / leaky tropical pipe?

Insofar as we have emphasized the importance of transport in the $O_3$-damped regime, it is a fair question whether transport other than the upward advection that we approximate as a damping might be important for ozone structure. To address this question, we have examined a version of the Chapman+2 model with explicit transport formulated as a leaky tropical pipe (instead of as $\kappa_{\overline{w}^*}$ in the damping of O and $O_3$). Our treatment builds on the analysis in Match and Gerber (2022). For our leaky tropical pipe, we adopt a three-column formulation with the tropics, NH extratropics, and SH extratropics, and we consider average upwelling of 0.3 mm s$^{-1}$ in the tropics, with compensating downwelling in the extratropics. The upwelling leads to mass divergence (leakage) and transport into the extratropics. We also consider lateral mixing on a timescale of 1 year (as in Stolarski et al. (2014)) and vertical diffusion of 0.01 m$^2$ s$^{-1}$ in the tropics and 0.5 m$^2$ s$^{-1}$ in the extratropics (as in Neu and Plumb (1999)). We impose zero ozone in the troposphere below 17 km.

In response to your comment, we address two main questions: (1) Does representing ozone transport with a leaky tropical pipe change the altitude of peak $[O_3]$ compared to in the Chapman+2 model where ozone transport is represented as a damping? (2) Does surgically replacing only the vertical advection in the leaky tropical pipe with damping by $\kappa_{\overline{w}^*}$, while retaining the lateral mixing and vertical diffusion, change the altitude of peak $[O_3]$?

The answer to the first question is "no": the Chapman+2 photochemical-transport model with a leaky tropical pipe has an interior maximum at 26.6 km, identical up to our O(1 km) level of approximation with the Chapman+2 model in which transport is represented as a damping.

The answer to the second question is also "no": when we set the vertical advection to zero in our leaky tropical pipe and replace its effects with $\kappa_{\overline{w}^*} = (3 \text{ months})^{-1}$ in the tropics (and no advection or damping in the extratropics), then the altitude of peak $[O_3]$ responds with only a negligible change of 0.2 km. The magnitude of ozone changes modestly, which is not surprising given that we have not attempted to optimize the magnitude of $\kappa_{\overline{w}^*}$ with respect to any particular benchmark other than our goal of an O(1 km) characterization of the altitude of the interior maximum. This response can be seen in Figure 1 of this Response to Reviewers.

To summarize, our approximation of transport as a linear damping reproduces the altitude of peak $[O_3]$ compared to if transport were represented with a more-realistic leaky tropical pipe. Our linear damping primarily represents the advective component of the transport, so simply replacing tropical advection in the leaky tropical pipe with our advection also does not significantly change the interior maximum of ozone.

[Figure]

Figure 1: Tropical $O_3$ profile for a Chapman+2 photochemical-transport model with transport by a leaky tropical pipe, including advection, leakage, lateral mixing, and vertical diffusion. We consider two ways of representing the advection and leakage: either accurately in their advective formulations (solid) or as approximated by a damping in the tropics of $\kappa_{\bar{w}^*} = (3 \text{ months})^{-1}$ and no treatment in the extratropics (dashed). Approximating the advective transport with a damping does not significantly affect the altitude of peak $[O_3]$, and it only modestly affects the magnitude in a way that could have been optimized with a different choice of $\kappa_{\bar{w}^*}$, but which we choose not to tune away because, for our purposes, the ozone magnitude only needs to be accurate enough not to alter the fundamental reason why ozone has an interior maximum through its effect on the photolysis rates.

L130 / R5 & R6. Why is it justified to treat vertical advection as an effective chemical loss? In particular, L133-134 imply a linear dependence of the generalized loss on the concentrations of O and O3. But advection by $\overline{\omega *}$ is proportional to the vertical gradient of the mixing ratio (equivalently, the appropriate term in the continuity equation), not to concentration. So, why does it work? This is actually discussed in Section 3.5.2 of Brasseur and Solomon 2005, but I think it should be briefly explained here too, especially that this approximate treatment of transport maybe responsible for some remaining inaccuracies of

the Chapman+2 model, as later explained in the discussion of Fig. 3.

We agree that approximating vertical advection as a linear damping does not generally work for arbitrary velocity and tracer fields. But, as you note, it does work in the case where ozone gradients are being smoothed by advection up from a low-ozone lower boundary condition (the tropopause). We like the discussion in Brasseur and Solomon, so we now cite that explicitly along with a modestly elaborated explanation in the text: "*Transport does not generally act as a linear damping, and indeed the Brewer-Dobson circulation is known to be a source of ozone in the extratropics (e.g., Dobson, 1956). However, in the tropical lower stratosphere, where transport might in principle be represented as a leaky tropical pipe (Neu and Plumb, 1999) such as in Match and Gerber (2022), in order to understanding peak $[O_3]$ its effects can be approximated as a linear damping. This linear damping results because ozone is being constantly upwelled from an ozone-poor region (the tropical tropopause layer) into a region over which it decays with a characteristic scale height (Brasseur and Solomon, 2005, Section 3.5.2). And, because transport is only important for ozone in the tropical lower stratosphere and not farther aloft (e.g., Garcia and Solomon, 1985; Perliski et al., 1989), a fact that will emerge self-consistently within the Chapman+2 model, parameterizing the effects of transport as a constant damping throughout the tropical stratosphere can lead to an accurate representation in the tropical lower stratosphere without imposing significant errors farther aloft. We consider that transport leads to a relaxation rate that scales with $\bar{w}^* = 0.3$ mm $s^{-1}$ divided by a reference vertical scale of approximately 2 km, leading to a damping rate of $\kappa_{\bar{w}^*} = (3$ months$)^{-1}$. For consistency, this damping will be applied to O and $O_3$, although it will be found to only significantly affect $O_3$ given the short lifetime of O.*"

Eqs. 5 and 6. I hesitate to ask for more math;) However, it may be instructive to see the full set of continuity equations from which these are derived under a steady state assumption. It could be another short appendix. It's just a suggestion. A committed reader should be able to back out the full set of equations from 5 and 6.

We share your general hesitation to add more equations, but the prognostic equations for [O] and $[O_3]$ do seem prudent to include, so we have added them to the main text as follows:

*These reactions can be incorporated into the Chapman cycle to yield a Chapman+2 model of tropical stratospheric ozone, with the following prognostic equations for O and $O_3$:*

$$\frac{\partial [\text{O}]}{\partial t} = 2 J_{\text{O}_2}[\text{O}_2] - k_2[\text{O}][\text{O}_2][\text{M}] + J_{\text{O}_3}[\text{O}_3] - k_4[\text{O}][\text{O}_3] - \kappa_{\text{O}}[\text{O}] \tag{1}$$

$$\frac{\partial [\text{O}_3]}{\partial t} = k_2[\text{O}][\text{O}_2][\text{M}] - J_{\text{O}_3}[\text{O}_3] - k_4[\text{O}][\text{O}_3] - \kappa_{\text{O}_3}[\text{O}_3] \tag{2}$$

L142. "implicit ... due to dependence on ozone aloft". So, not really quadratic in $[O_3]$ to leading order as stated earlier in the same sentence? Or is the dependence of the

photolysis rates on integrated ozone sub-quadratic/weak? Is any of this relevant to the calculations?

We have clarified that the equations are mathematically implicit *in height*, which is quite important to solving them, as one must integrate from the top of the atmosphere downwards. They are quadratic at a given altitude. We now say, *This equation is quadratic in [$O_3$] and mathematically implicit in height due to the dependence of $J_{O_2}$ and $J_{O_3}$ on ozone aloft.*

LL156-157. See my comment to L130 above.

We have addressed the justification for approximation the effects of upwelling as a linear damping the tropical lower stratosphere above.

L186. I would imagine that the diurnal cycle at those altitudes is pretty much negligible compared to the other factors.

We believe that all of these approximations are acceptable for reproducing the altitude of peak [$O_3$] with O(1 km) accuracy. It is not obvious to us why the diurnal cycle of solar zenith angle should necessarily have a much smaller effect than these other factors that we ignore, so in the interest of humility, we will continue to note its omission as a caveat.

L197-199 and 210. Nice! Thanks!

L 202. "Chapman dynamics"? Or chemistry? Fair point about semantic ambiguity as to whether "dynamics" refers to transport or the broader behavior of a dynamical system. We now avoid this ambiguity here by saying "*under Chapman photochemistry*".

Eq 11. Shouldn't there be a factor of $n_a$ in the numerator? Equation 5 has an $n_a^3$ in the prefactor and $n_a^2$ in the relevant term in the denominator. The original equation is correct. The factor of $n_a$ that you expect to see is included in the number density of $O_2$, where [$O_2$] $= C_{O_2} n_a$.

L242. Typo: duplicate "regimes" Corrected.

L247. As above. Is this a climatological mean tropical profile from MERRA-2? A profile for a specific day? See response above regarding our switch to using SWOOSH data.

L250-251, Fig 5. I suggest cutting the figure off at the approximate altitude of the tropopause. Everything below that is irrelevant to this discussion. We have adopted this convention for all relevant figures in the paper, which are now all cut off at 15 km (with the exception of a couple schematic illustrations). We agree that tropospheric ozone is largely irrelevant to this discussion, and that we are instead primarily interested in understanding stratospheric ozone and its interior maximum.

L271 and below. Is "non-analytic" used in the mathematical sense of not having a Taylor expansion or in the loose sense (no closed formula)? In the latter case I think people say "non-analytical". I may be wrong.

We have clarified our language, where we did not mean to refer to a "non-analytic function" but rather to a "*non-analytical function*". Thank you for pointing this out.

L351, 354. Since Eq. 23 contains the Lambert W-function it may be questioned whether this qualifies as an analytical solution. I'm not sure if there's a closed formula

for it even in this case where the argument is real and positive. As far as I know W is evaluated numerically using rather terrifying-looking approximations (e.g. `https://mathworld.wolfram.com/LambertW-Function.html`).

We understand that it is appropriate to include certain special functions in an expression and still call it analytical, although it might be incorrect had we chosen to refer to such an expression as "closed-form" or "exact". In the Wikipedia entry for "closed-form expression", it is noted that the term "analytic expression" (what our community colloquially calls analytical expressions) "tends to be wider than that for closed-form expressions. In particular, special functions such as the Bessel functions and the gamma function are usually allowed, and often so are infinite series and continued fractions."

L390. The profiles in Fig. 7 look continuous to me. What am I missing?

This comment refers to the discontinuity at the regime transition in the analytical expression under gray radiation (gray dashed curve). All of the other profiles are continuous.

L543. I think it's supposed to be Appendix C. Corrected.

**1.2 References**

Gelaro, R., McCarty, W., Surez, M. J., Todling, R., Molod, A., Takacs, L., et al. (2017). The Modern-Era Retrospective analysis for research and applications, version 2 (MERRA-2). Journal of Climate, 30(14), 5419-5454. `https://doi.org/10.1175/JCLI-D-16-0758.1`

Wargan, K., Labow, G., Frith, S., Pawson, S., Livesey, N., & Partyka, G. (2017). Evaluation of the ozone fields in NASA?s MERRA-2 reanalysis. Journal of Climate, 30(8), 2961-2988. `https://doi.org/10.1175/JCLI-D-16-0699.1`

Thank you again for your thoughtful and supportive review, Kris! Your comments have improved the paper.

**2 Review 2**

**Review of "Protection without poison: Why tropical ozone maximizes in the interior of the atmosphere" by A. Match et al.**

**2.1 General**

I think it is good that this paper addresses issues of catalytic cycles of ozone loss and ozone production in the tropical stratosphere. The study goes back to textbooks and examines what can be learned from the 'classic' explanations and in how far they should perhaps be modified.

However, the paper needs to have a clearer message. I do not think that the fact that tropical ozone has a peak (in concentration) at about 26 km is the new finding that this papers wants to report. And also not the concentration of ozone at this altitude. As I read

the paper, what is new here is the advance in understanding of several issues that is gained through the different simplified models. I do not think this aspect is coming across very well.

The analysis in the paper also based on certain assumptions, which are important for the presented derivations. I suggest mentioning these assumptions clearer and upfront (rather than in the course of particular derivations). There are also questions about the "real world" ozone representation (i.e., MERRA-2, see also below).

Overall I suggest improvements to the manuscript. The message of the paper should come across more clearly.

We thank the reviewer for their constructive comments. In response to this review, we have attempted to broadly clarify the paper in several ways. Most importantly, it seems that the introduction was not as effective as it could be in preparing the reader to understand our contribution. We have sharpened the introduction by improving Figure 2 to contrast the ozone scaling from the source-controlled paradigm with the ozone scaling from the source/sink competition paradigm, and we show how these scalings are each biased in their altitude of peak $[O_3]$ by $O(10 \text{ km})$. We then clearly state our goal and some key associated assumptions and geographical restrictions: "*We seek a minimal, steady-state theory for the tropical stratospheric $[O_3]$ maximum that invokes realistic sinks from catalytic cycles and transport and yields a prediction for the interior maximum of ozone that is accurate to approximately 1 km.*" We have clarified up front that our theory is steady-state and restricted to the tropics, both concerns of this reviewer.

The source of our ozone data is ultimately not important to this argument, because any data source over the past 50 years could suitably indicate that the ozone maximum is around 26 km. For example, in the textbook of Dobson (1963) cited by this reviewer, Dobson writes that the ozone maximum is between 25 km and 27 km. This is already within the level of approximation sought by our explanatory framework, which was implicit but is now clearly stated as $O(1 \text{ km})$. Thus, our primary goal with sourcing the ozone data is not to distract the reader from the main point that ozone is well known to maximize around 26 km. Towards that end, we have switched to using observational data from SWOOSH, which is a homogenized dataset of satellite data since 1984.

Responding to the concerns of both reviewers, we have restricted our plots to omit the troposphere, which is not relevant to our argumentation and is not represented in our modeling framework. This further clarifies the paper as being focused on the stratospheric photochemical-transport regime

We have clarified some of our language, opting for more distinct and evocative terms when possible. Instead of making the central dichotomy of the paper the distinction between the photolytic sink regime and the non-photolytic sink regime, which is accurate but perhaps too abstract, we have made it the distinction between the O-damped regime the $O_3$-damped regime, which we then note can be generalized into a distinction between a photolytic sink regime and non-photolytic sink regime. Whereas we previously referred to gray radiation and monochromatic radiation interchangeably, we now refer only to gray

radiation.

**2.2   Comments**

**Applicability of the analysis**   Clearly, the maximum (in concentration) of ozone is not the same everywhere, in particular it depends on latitude. It is not clear which region the paper is addressing, looking at section 2 it seems the entire atmosphere (but the the maximum is more than one value); however the title says 'tropical'. Further, the maximum of ozone concentration at a particular altitude (even for a given latitude) can only be considered a fixed number for some kind of climatology (i.e., for some averaging; see also below). This aspect should be addressed in the paper, e.g., there should be more information in the caption of Fig. 2 (which latitude range, which period? etc., see also comments on Fig. 3). Transport is represented rather crudely (l. 120- 127) and the analysis is restricted to the tropical lower stratosphere. So is the tropical (lower) stratosphere the regime to which this analysis should be applied? I think the answer is 'yes', but this should be clearly stated throughout the paper.

Our title does state that our paper is about "tropical ozone", and we state this restriction numerous times. However, given that it is such an important restriction on the scope of our study, we have further propagated our restricted focus on the tropics into our explanations of our goals and results. For example, we now state the central goal of the paper as follows: "We seek a minimal, steady-state theory for the tropical stratospheric $[O_3]$ maximum that invokes realistic sinks from catalytic cycles and transport and yields a prediction for the interior maximum of ozone that is accurate to approximately 1 km."

Our data source has been switched to SWOOSH, now introduced early in the Introduction as follows: "*The tropical stratospheric peak in $[O_3]$ is robust across observational datasets. As an observational benchmark, this paper uses the homogenized satellite dataset SWOOSH (Davis et al., 2016), averaged over the tropics (30° S-30° N) and from 1984-2023. In SWOOSH, monthly tropical $[O_3]$ peaks at 26 km, deviating only about 10% of the time up or down from this altitude by at most one vertical level of roughly 1 km.*"

Our representation of transport is tailored to the tropical lower stratosphere but can be imposed without inducing undue error throughout the entire tropical stratosphere due to the transition from an $O_3$-damped regime below to an O-damped regime aloft. This applicability to the tropical stratosphere is now stated as follows: "*And, because transport is only important for ozone in the tropical lower stratosphere and not farther aloft (e.g., Garcia and Solomon, 1985; Perliski et al., 1989), a fact that will emerge self-consistently within the Chapman+2 model, parameterizing the effects of transport as a constant damping throughout the tropical stratosphere can lead to an accurate representation in the tropical lower stratosphere without imposing significant errors farther aloft.*"

Another aspect is tropospheric ozone. At some instances in the paper it is clearly stated that this analysis is not about tropospheric ozone chemistry (l. 468). This is correct and I agree. But this is not clearly and upfront stated in the paper. And some of the figures

(e.g. 2, 4, 5) extend to the ground, leaving the impression that the discussion in the paper is relevant also for tropospheric altitudes (for which is it not). Looking at Figs. like Fig. 5, it is clear that the real atmosphere (here MERRA-2, see also below) does not look like anything in the 'non-photolytic sink regime' below the tropopause. Moreover, given the fact that the tropical tropopause is at about 18 km (Hoffmann and Spang, 2022); most of the the 'non-photolytic sink regime' in Fig. 5 is irrelevant.

In response to this recommendation from both reviewers, we have cut off most of our figures below 15 km. This avoids distracting the reader with the ozone profile of the troposphere, which is largely irrelevant to our main arguments. Thank you for this comment, which improves the paper. Our main goal is now clearly stated as explaining the *stratospheric* [O$_3$] maximum, an understanding of which is sufficient to explain the interior maximum in the tropics.

Given our goal of explaining the tropical stratospheric ozone maximum to within approximately 1 km, the choice of dataset is not strongly relevant to this problem, because the datasets agree to within approximately 1 km. The altitude of the ozone maximum is robust in time and across datasets. So, our primary goal should be to not distract readers with any handling of data that appears to be unwarranted or arbitrary. It is understandable why, to this reviewer, our use of an atmospheric reanalysis and consideration of only the representative year of 2018 both appeared unwarranted and arbitrary, so we have opted instead to consider observations across the satellite record since 1984 by using the SWOOSH dataset of homogenized satellite observations. The time-averaged ozone maximum is at 26 km. Also, the monthly-varying ozone maximum is at 26 km almost 90% of the time. Thus, the tropical [O$_3$] maximum at 26 km is a robust feature of the climate system, and its internal variability is of the same order as the accuracy we seek in our theory. About the interior maximum in SWOOSH, we now state the following: "*The tropical stratospheric peak in [O$_3$] is robust across observational datasets. As an observational benchmark, this paper uses the homogenized satellite dataset SWOOSH (Davis et al., 2016), averaged over the tropics (30° S-30° N) and from 1984-2023. In SWOOSH, monthly tropical [O$_3$] peaks at 26 km, deviating only about 10% of the time up or down from this altitude by at most one vertical level of roughly 1 km.*"

**Damping** Throughout the paper, the concept of "damping" is used. It is mainly applied to ozone (O3) or to O (where Ox = O + O3). To me "damping" is the energy loss of an oscillating system through dissipation. Here this concept seems to be used as a synonym for change of O3 or O at a particular location through chemical (catalytic) loss and advection of O3 (e.g., advection of low ozone from below in the tropical lower stratosphere).

The concepts of chemical change and advection of stratospheric ozone are well known (see e.g. the textbooks cited in this manuscript), so I do not really see the advantage of redefining these concepts in terms of "damping". At least, I suggest that the wording used here is related to the well established concepts of catalytic loss and advection of stratospheric ozone. Furthermore, these concepts apply to ozone in photochemical equilibrium,

where $\partial O_3/\partial t = \partial O/\partial t = 0$— correct?

We refer to damping in the sense of a term in a dynamical system of the form $\partial X/\partial t = -\kappa X + ...$ that reduces variability, with damping rate $\kappa$. This dynamical sense of the term is frequently used in atmospheric science, where damping can refer to Newtonian damping (also known as Newtonian cooling) in the thermodynamic energy equation ($\partial T'/\partial t = -\kappa T' + ...$) or Rayleigh damping (also known as Rayleigh friction) in the momentum equation ($\partial u/\partial t = -\kappa u + ...$) (e.g., Fels, 1982; Andrews et al., 1987; Romps, 2014). Such damping can be useful when studying oscillatory solutions, as noted by the reviewer, but can also be useful when characterizing steady states or the freely-evolving dynamical evolution of a complex system (e.g., Gill, 1980; Held and Suarez, 1994). We are aware that in chemistry parlance, a linear damping term of this form is referred to as a "first-order decomposition reaction" or a "unimolecular decomposition reaction", and we now inform readers of this equivalence.

We have hopefully clarified our terminology and some its attendant assumptions up front in the section introducing our Chapman+2 model. Our revised sub-section now describes our approach as follows:

*"Neither transport nor catalytic cycles generally act as a linear damping in all parts of the atmosphere or in all photochemical regimes. However, we will argue that the tropical stratosphere is in a regime where they can be fruitfully parameterized as such, facilitating theoretical insight.*

*Transport does not generally act as a linear damping, and indeed the Brewer-Dobson circulation is known to be a source of ozone in the extratropics (e.g., Dobson, 1956). However, in the tropical lower stratosphere, where transport might in principle be represented as a leaky tropical pipe (Neu and Plumb, 1999) such as in Match and Gerber (2022), in order to understanding peak $[O_3]$ its effects can be approximated as a linear damping. This linear damping results because ozone is being constantly upwelled from an ozone-poor region (the tropical tropopause layer) into a region over which it decays with a characteristic scale height (Brasseur and Solomon, 2005, Section 3.5.2). And, because transport is only important for ozone in the tropical lower stratosphere and not farther aloft (e.g., Garcia and Solomon, 1985; Perliski et al., 1989), a fact that will emerge self-consistently within the Chapman+2 model, parameterizing the effects of transport as a constant damping throughout the tropical stratosphere can lead to an accurate representation in the tropical lower stratosphere without imposing significant errors farther aloft. We consider that transport leads to a relaxation rate that scales with $\bar{w}^* = 0.3$ mm $s^{-1}$ divided by a reference vertical scale of approximately 2 km, leading to a damping rate of $\kappa_{\bar{w}^*} = (3 \text{ months})^{-1}$. For consistency, this damping will be applied to O and $O_3$, although it will be found to only significantly affect $O_3$ given the short lifetime of O.*

*Like transport, catalytic cycles also do not generally act as a linear damping. This is because they involve two- (and sometimes three-)body reactions whose rates depend on the abundance of the catalysts, which are often co-evolving with the overall photochemical state. Thus, in order to treat catalytic destruction of ozone as a linear damping with a steady,*

*altitude-dependent damping rate, we assume that the number density of the catalysts and the temperature-dependent reaction rates are constant. We then must use these constant profiles of damping rates to damp either O or $O_3$."*

**Twenty six kilometres** Throughout this paper (starting with the abstract, ls. 2, 10) the altitude of the peak of the ozone layer (in terms of concentration) is a key point here. And the transition between two photochemical regimes. Twenty six kilometres is the number that is reported. However, how certain is this number; no error estimate is given, what is the uncertainty for the 26 km? Also, is there a variability of the 26 km with other atmospheric parameters (e.g. QBO, Nivano et al., 2003; Diallo et al., 2018).

Perhaps more importantly, any successful simulation of the tropical ozone layer should not only reproduce the location of the peak in altitude but also the vertical distribution in ozone concentration with altitude (including the ozone concentrations at the peak); I think this aspect could be treated more thoroughly in this paper.

The interior maximum of tropical $[O_3]$ is robust in time and across datasets. Here is what we now say about that: "'*The tropical stratospheric peak in $[O_3]$ is robust across observational datasets. As an observational benchmark, this paper uses the homogenized satellite dataset SWOOSH (Davis et al., 2016), averaged over the tropics (30°S-30°N) and from 1984-2023. In SWOOSH, monthly tropical $[O_3]$ peaks at 26 km, deviating only about 10% of the time up or down from this altitude by at most one vertical level of roughly 1 km.* "

Our focus on the paper is about understanding the regime transition, but the paper does also offer a theory for the vertical profile of tropical stratosphere ozone. The magnitudes of ozone in this theory must be accurate enough only to the extent that they do not lead to unrealistic photolysis rates that change the fundamental reason why there is an interior maximum of ozone, an error that does in fact occur in the Chapman cycle, which overestimates ozone by a factor of two. We treat this idea in the Discussion when we explain why the Chapman cycle simulates the correct altitude of ozone for the wrong reason, because its biases in ozone magnitude fortuitously correct the erroneous magnitude that would have otherwise been predicted in the source/sink competition paradigm. The Chapman+2 model corrects these factor-of-2 errors, and so is able to reproduce the interior maximum of ozone for the correct reason.

**Assumptions** There are a number of assumptions on ozone made in this study that are mentioned; however I suggest to state these assumptions more clearly ad upfront. For example, photochemical equilibrium of ozone is assumed ($\partial[O_3]/\partial t = 0$; l. 138), which is a strong assumption. This assumption is not valid for a large part to the stratosphere. (Which seems to be the reason why this analysis is restricted to the tropics). Further there are assumptions like an isothermal atmosphere (also not realistic) and other assumptions (l. 293) that should be clear.

Referring to the steady state of the Chapman+2 model as photochemical equilibrium was a confusing choice of words on our part, because it suggested that we were neglecting transport when in fact the effects of transport are parameterized through the damping, which analogizes the effects of transport to that of a first-order decomposition reaction. Now, instead of referring to the resulting equilibrium as a photochemical equilibrium, which we realize typically excludes the effects of transport, we now refer to such an equilibrium as a "steady-state". The statement of this steady-state assumption has been moved earlier in the paper, including into the Introduction. For example, we now state in the Intro: "*We seek a minimal, steady-state theory for the tropical stratospheric [O$_3$] maximum that invokes realistic sinks from catalytic cycles and transport and yields a prediction for the interior maximum of ozone that is accurate to approximately 1 km.*"

We now provide a condensed overview of some of our key assumptions when evaluating the Chapman+2 model: "*Agreement between the Chapman+2 model and observations is imperfect, which is unsurprising given that this work employs many simplifying approximations. We have assumed overhead sun impinging on an isothermal atmosphere, approximated transport and catalytic cycles as a linear damping, used globally-averaged catalytic profiles, and neglected optical scattering. All of these approximations (and more) will be necessary later when we derive an explicit analytical expression to the Chapman+2 model ozone profile. Despite these approximations, the Chapman+2 model produces a reasonable fit to the observed profile, and will be considered to produce a credible interior maximum of ozone. The remainder of the paper seeks to explain why the Chapman+2 model produces an interior maximum.*"

**Catalytic cycles destroying ozone**    The Chapman (1930) model is known to be incorrect insofar as it neglects the most relevant catalytic ozone loss cycles (e.g., Portmann et al., 2012). Thus to investigate tropical ozone (which seems to be the target here) one needs to look at the tropical profiles for the relevant species driving ozone loss in the tropics. I do not think this is the case here and I recommend changing it. Global estimates are not helpful here, the atmosphere is very different in the tropics and in the mid- latitudes. Already decades ago, researchers (including the editor of the present manuscript) have invested substantial effort into deriving ozone loss cycles in the tropics (e.g., Crutzen et al., 1995).

We are aware that it is an assumption to approximate the tropical catalysts profiles with those from the global average. This assumption would not be suitable for certain types of problems. However, for our problem of reproducing the interior maximum of ozone up to an accuracy of O(1 km), it appears to be post hoc acceptable, given the success of the Chapman+2 model when driven by realistic boundary conditions.

We have tested this sensitivity to latitude and also to source model by comparing our Chapman+2 model solution with globally-averaged catalysts from SOCRATES to an experiment using tropically-averaged catalysts from the chemistry-climate model MRI-ESM2-0. Not all catalysts were available in the publicly accessible output of MRI-ESM2-0,

but the most important ones were available, so we replaced them with the MRI-ESM2-0 profile and retained the SOCRATES profiles for the others. The resulting effects on the ozone profile could be important for some problems, but are negligible for our purposes. Figure 2 of this Response to Reviewers shows the result. They do not strongly change the ozone concentration beyond the level of other approximations we have made, and they only change the altitude of the ozone maximum by 0.2 km, which is within our intended accuracy of O(1 km). Thus, we will retain the globally-averaged profiles, noting this approximation as one among many in the simple framework of this paper.

[Figure]

Figure 2: (left) Fractional change in the catalyst profiles when moving from the globally-averaged SOCRATES data to tropically-averaged MRI-ESM2-0 data. (right) Resulting Chapman+2 model experiments in a Control configuration using the globally-averaged SOCRATES catalysts compared to a Modified configuration in which the profiles of $NO_2$, $HO_2$ and OH were switched to the tropically-averaged MRI-ESM2-0 data while retaining other catalysts from globally-averaged SOCRATES. The sensitivity of the ozone profile is tolerably within the level of approximation targeted by our simple theory.

**Formation and destruction of ozone**  In the introduction, there is a discussion of the ozone sources and sinks in the atmosphere. While I like the idea to go back to the textbooks, it should be clear that much more is known today than what is discussed in Fig. 1. Some modern textbooks are cited, but another example is Portmann et al. (2012). Regarding the history of the debate on the relevant sinks of ozone in the atmosphere I also recommend the book by Brasseur (2019).

We also quite like the historic perspective of Brasseur (2019) and the study of anthropogenic perturbations to the ozone layer by Portmann et al. (2012). We believe the reviewer is concerned that our emphasis on the Chapman cycle in textbook explanations is misaligned with the general understanding encoded in those same textbooks that the Chapman cycle omits leading-order sinks of ozone. This presents a pedagogical challenge in the textbooks, and might be part of why the explanatory landscape has fragmented, as we note between the two leading paradigms. In order to preemptively assure the reader that the Chapman cycle is never considered to capture the dominant sinks of ozone in any of the textbooks surveyed, we now note a useful caveat as follows: "*We surveyed ten atmospheric radiation and chemistry textbooks and found that, when explaining the structure of the ozone layer, all ten introduce the Chapman cycle, even as many also note its important omissions of catalytic cycles and transport.*"

**Odd oxygen** Commonly, odd oxygen Ox is defined as Ox = O + O3. This is also done here. This is a well established concept (see e.g. the textbooks cited in this manuscript). However, Ox is used here first (l. 132) before being defined (l. 136). But more importantly, the concept needs to be introduced briefly before being used (even if the reviewer has learned about Ox before). Further, below (l. 163) Oy is mentioned — but without knowing what this is, any discussion below on Oy is not very useful. Define in this paper what Oy is.

Odd oxygen is now defined upon its first use as follows: "*Whether the damping is primarily of O versus of $O_3$ will turn out to lead to qualitatively different mechanisms for ozone structure, a surprising result given that O and $O_3$ are often treated as conceptually fungible within the chemical family of odd oxygen ($O_x \equiv O + O_3$) (Section 2).*"

Generalized odd oxygen $O_y$ is used in one paragraph to make a link between our Chapman+2 model and the formalism of Brasseur and Solomon (2005). That discussion now more clearly refers the reader to the textbook should they desire a deeper dive: "*The catalytic component of these damping rates can be related to the budget of generalized odd oxygen ($O_y$), which was defined in Brasseur and Solomon (2005) to include a broader set of chemical constituents that can serve as reservoirs for odd oxygen under stratospheric photochemistry. Equations … include all of the dominant sinks of $O_y$ that are linear in O or $O_3$. These damping rates treat the concentrations of catalysts as constant and neglect conversions of generalized odd oxygen between reservoir species, so do not provide an exhaustive account of the $O_y$ budget. Nonetheless, they will serve to effectively parameterize the sinks of O and $O_3$.*"

**Ozone in the real world** As a reference for ozone in the real atmosphere, MERRA-2 (Gelaro et al., 2017) is used. Always one particular profile is presented , but it is not clear what the profile shows. In the caption of Fig. 3, one reads that this is a tropical ozone profile (30S to 30N), but this is not the most conservative estimate of the tropics; perhaps

20- 20 would be more appropriate? Has this sensitivity been explored? The profile is for 2018. I could not find another place in the manuscript where the profile is explained (which should be changed). I assume that it is a zonal mean profile and that the profile is annually averaged (of course, I am not sure). But this should be clearly stated in the manuscript. Another question is, which vertical resolution was used for MERRA-2 ozone.

Further, how was 2018 chosen? As pointed out in the manuscript tropical ozone depends on the solar UV-flux, which is changing with the 11-year solar cycle (albeit not by a factor of two). Tropical ozone might also be influenced by the QBO (see above); could this point be of interest here? If only one single year is considered, why was a reanalysis chosen rather than direct measurements (e.g. MLS, Waters et al., 2006; Han et al., 2019). A climatology based on (ozone) observations could be an alternative (e.g., SWOOSH, Davis et al., 2016).

We have previously addressed the issue of the reference profile by switching to SWOOSH and stating up front our averaging procedure. We additionally note here that the interior maximum of ozone is not significantly sensitive to the averaging window within the tropics. When we define the window from 20S-20N, the interior maximum is still at 26 km, and is furthermore at that altitude during 80% of the individual months from 1984-2023.

**Technical issues** The paper could be easier to read. There are a few issues where a clearer language would help. The paper talks about a "gray ozone layer" — to me this is jargon. This term (gray ozone) is not familiar to the readers of ACP; it is rather a shorthand for assumptions about the lack of a spectral dependence of $\sigma_{O_3}$ and $\sigma_{O_2}$.

We attempted to clarify some of the language along these lines. We have replaced all instances "gray ozone layer" with "ozone layer under gray radiation" or similar. Having previously referred to gray radiation interchangeably as monochromatic, we now only refer to gray. Having previously emphasized the dichotomy between the fairly abstract concepts of the photolytic sink regime and non-photolytic sink regime, we now refer to the dichotomy between the more evocative O-damped limit and $O_3$-damped limit. We hope that these and other revisions have improved the readability, and thank the reviewer for encouraging us to consider such edits.

In Eq. 7 (l. 160) $a_5$ (and similar coefficients) are used. I cannot see that these coefficients have been defined before (did I overlook anything?). It would be good to explain here what the atmospheric meaning of $a_5$ (etc.) is.

We now clarify that these are reaction rate coefficients: "*we have adopted the variable names for the reaction rate coefficients $a_5$, $a_7$, $b_3$, etc. from Brasseur and Solomon (2005)*".

Also, $O_y$, is used in the paper, but this is not an established notation (see also the comments on odd oxygen above). I also suggest a better introduction to the concept of 'damping' to make the paper more easily accessible (again see discussion above). Further, there is 'appendix B' two times in the paper — this is confusing.

These issues have been addressed/corrected in previous comments.

There are recommendations by ACP; the abstract is likely too long. Also, it would be good if the coefficients for the Cariolle 2.9 scheme used here were available — in this

way the results of this paper could be reproduced and the coefficients be used for other purposes.

The abstract is presently <250 words, and we await further instruction from the editor if it is too long.

We are sympathetic to the reviewer's intentions that the Cariolle coefficients, which play a minor but useful role in validating our approach in this paper, were publicly accessible. Unfortunately, those coefficients are not our intellectual property and it would be inappropriate to publish them without consent. We have informed Cariolle and his collaborator, who were happy to share the coefficients by personal communication with us, of the potential value of making this dataset public. We said, "A point that arose during peer review is that it would be valuable for the ozone community to have a publicly accessible version of these coefficients. I declined to publish my own copy of the coefficients because they are your intellectual property, but I think that a publicly accessible version could increase the use of this highly valuable dataset."

**Minor issues**

- l. 17: This is a matter of opinion, but I suggest not attributing the discovery of the ozone layer to Hartley alone; see the discussion by Brasseur (2019).

  We enjoy the historic perspective of Brasseur (2019) and his careful documentation of the evolving understanding of the ozone layer through the numerous ground-breaking contributions of not only Hartley but also Fabry and Buisson, and others. That said, for the purposes of introducing the concept of protection from UV, we are comfortable attributing the discovery to Hartley, who concluded his 1881 paper as follows:

  "The foregoing experiments and considerations have led me to the following conclusions:

  1st. That ozone is a normal constituent of the higher atmosphere.

  2nd. That it is in larger proportion there than near the earth's surface.

  3rd. That the quantity of atmospheric ozone is quite sufficient to account for the limitation of the solar spectrum in the ultra-violet region, without taking into account the possible absorption caused by the great thickness of oxygen and nitrogen."

- l. 35: I suggest also to have a look at the classic textbook by Dobson (1963). While the textbooks want to give a simple message to the reader, it is clear that production of ozone alone, without a loss mechanism for ozone would simply convert O2 to ozone. This is why Dobson (1963), on page 105 of his book, calls the chapter "FORMATION AND DESTRUCTION OF OZONE".

  We have studied Dobson's 1963 textbook explanation of ozone layer structure, and it is consistent with subsequent textbook treatments like those considered in our paper. In particular, Dobson's treatment is precisely in line with that of the standard (albeit

less common) source/sink competition paradigm, which is ultimately based on the Chapman cycle. The sink of ozone that he considers is from the reaction of O and $O_3$.

Perhaps the reviewer is concerned that the source-controlled paradigm does not invoke an explicit sink. We would note that just because it does not invoke an explicit sink does not mean that they imply that ozone has no sink and would therefore build up indefinitely. This point is not so clear in some textbook explanations, but a generous interpretation is that the source-controlled paradigm means that the sink of ozone is not interpreted to control the shape of the ozone layer. Such a passive sink is exactly what results in the $O_3$-damped regime under uniform $\kappa_{O_3}$, in which case the structure of the ozone layer is dictated by the shape of the ozone source.

- l. 50: The quoted MERRA-2 ozone profile is for the tropics (information only in the caption). But how are the tropics defined here? For which period is the ozone profile valid? Likely the MERRA-2 ozone is the same as the one in Fig. 3 — correct? Does MERRA-2 assimilates ozone observations? Perhaps add a reference for MERRA-2 (see above).

  As noted above, we have switched to SWOOSH and clearly defined the tropics as 30S-30N, a definition to which the peak $[O_3]$ is not unduly sensitive.

- l. 64: Is is clear here what passive and active sinks are?

  For clarity, we have omitted discussion of passive and active sinks.

- l. 83: 'gray radiative transfer' is not clear to me here (see also above).

  For clarity, we now refer to the "ozone layer under gray radiation".

- l. 83: 'endogenously' sounds like a medical term to me — is it really helpful here?

  For clarity, we have switched "endogenously" to "self-consistently".

- eq. 3: larger brackets for the exponential function (also elsewhere)

  For readability, larger brackets are now used for fractions inside parentheses here and elsewhere.

- l. 122: 'damping ozone' could be better explained

  For clarity, we have overhauled our introduction of the Chapman+2 model, including how we introduce the concept of damping. Relevant to this comment is this revised text: "*The Chapman cycle neglects the dominant sinks of ozone from catalytic chemistry and transport (Bates and Nicolet, 1950; Crutzen, 1970; Jacob, 1999; Brasseur and Solomon, 2005). These sinks involve photochemical reactions and transport among a system of at least tens of significant constituents. The consequences of the additional sinks of ozone from these processes is that ozone is approximately*

*halved compared to in the Chapman cycle. Thus, calculating accurate photolysis rates, which depend on overhead column ozone, requires an accurate representation of basic state ozone as it is affected by these sinks. But, while the effects of these sinks are essential, many of their details are not thought to be part of a minimum essential explanation for the interior maximum of ozone. Therefore, we will parameterize the effects of these sinks on $O$ and $O_3$, facilitating a simple and tractable theory with a realistic basic state ozone profile. These sinks are parameterized by augmenting the Chapman cycle with two linear damping reactions that destroy $O$ and $O_3$ respectively:*

*Representing these sinks as a linear damping is equivalent to adding an extra sink of $O$ and $O_3$ in the form of a first-order decomposition reaction (analogous to radioactive decay). "*

- l. 128: is this 'augmentation' meant to be globally or tropical?

  For clarity, we have stated, starting with the title, that our goal is to explain tropical ozone.

- l. 130: I think the coefficients zO and zO3 are important for this paper, but they are not reported (did I miss anything?)

  The coefficients are functions of height that are plotted in Fig. 3c and are calculated following the equations stated in the paper as part of the freely available repository of code accompanying this paper and reported as follows in the Code and Data Availability statement: "*The Chapman cycle Photochemical Equilibrium Solver described in Section 2 is published at doi: 10. 5281/ zenodo. 10515739.*"

- Fig. 3: I would not call MOBIDIC a chemistry climate model; I think nowadays something else is understood by the term chemistry climate model.

  Thanks. We now refer to MOBIDIC as a chemical transport model.

- l. 157: how is this vertical scale determined?

  We have sharpened our explanation of the representation of transport in the Chapman+2 model.

  "*Transport does not generally act as a linear damping, and indeed the Brewer-Dobson circulation is known to be a source of ozone in the extratropics (e.g., Dobson, 1956). However, in the tropical lower stratosphere, where transport might in principle be represented as a leaky tropical pipe (Neu and Plumb, 1999) such as in Match and Gerber (2022), in order to understanding peak $[O_3]$ its effects can be approximated as a linear damping. This linear damping results because ozone is being constantly upwelled from an ozone-poor region (the tropical tropopause layer) into a region over which it decays with a characteristic scale height (Brasseur and Solomon, 2005, Section 3.5.2). And, because transport is only important for ozone in the tropical lower stratosphere and*

*not farther aloft (e.g., Garcia and Solomon, 1985; Perliski et al., 1989), a fact that will emerge self-consistently within the Chapman+2 model, parameterizing the effects of transport as a constant damping throughout the tropical stratosphere can lead to an accurate representation in the tropical lower stratosphere without imposing significant errors farther aloft. We consider that transport leads to a relaxation rate that scales with $\bar{w}^* = 0.3$ mm $s^{-1}$ divided by a reference vertical scale of approximately 2 km, leading to a damping rate of $\kappa_{\bar{w}^*} = (3$ months$)^{-1}$. For consistency, this damping will be applied to O and $O_3$, although it will be found to only significantly affect $O_3$ given the short lifetime of O.*"

Formally, the vertical scale of approximately 2 km is not exogenous to the problem of predicting ozone structure in the tropics, but rather encodes the fact that ozone tends to increase in the tropical lower stratosphere with a vertical scale of $O(2$ km$)$. Thus, an even more unsparing first-principles approach to modeling ozone would need to estimate this height scale some other way.

- l. 167: I do not understand, why 'globally averaged' profiles of chemical constituents are used. The paper is on tropical ozone.

  As discussed above, globally-averaged profiles are used because they are accessible to us and the readers through Brasseur and Solomon (2005), and we have shown that using tropically-averaged profiles for a subset of the most important catalysts from MRI-ESM2-0 does not cause errors beyond our tolerance of explaining the ozone maximum to $O(1$ km$)$.

- l. 176 'eff' should not be in italics

  Corrected.

- l. 182: more need to be explained here that 'average' — see also above.

  As previously addressed, the observational ozone profile used throughout the paper is now introduced and clearly stated to be used throughout: "*The tropical stratospheric peak in $[O_3]$ is robust across observational datasets. As an observational benchmark, this paper uses the homogenized satellite dataset SWOOSH (Davis et al., 2016), averaged over the tropics ($30°S$-$30°N$) and from 1984-2023. In SWOOSH, monthly tropical $[O_3]$ peaks at 26 km, deviating only about 10% of the time up or down from this altitude by at most one vertical level of roughly 1 km.*"

- l. 187: I do not think that it is necessary to use globally averaged profiles here. (Also the profile is probably not 'catalytic'.)

  Addressed above.

- l. 197: Does the Chapman cycle sink ever dominate in the atmosphere?

No, the Chapman cycle sink is never dominant, at least in the stratospheric regimes of interest. This is consistent with much previous work including the discussion in Brasseur and Solomon (2005), where the Chapman cycle loss of ozone is never appreciably greater than 10-20% of the total loss (their Figure 5.71).

- l. 216: adding the reactions that lead to the 'domination' would be helpful here.

  We agree that is useful to the reader and have stated which reactions dominate in each regime.

- Fig. 4: It should be clear that this figure is for the tropics. Second, the figure extends to the ground, but the tropospheric chemistry prevailing below $\approx$ 18 km is not discussed here.

  For clarity, we have added "*tropics*" to virtually all of our figures where relevant. Per this recommendation and as previously discussed, we have truncated figures at 15 km.

- l. 241: It is nor clear here where the 26 km value comes from.

  This has been addressed above.

- l. 270: solutions are only for equations, not for a 'layer'. (Also l. 294).

  Revised to "*We preface our derivation of an explicit expression for the profile of ozone under idealized boundary conditions by first noting that there are no previously published mathematically explicit expressions for the ozone profile under any set of assumptions, let alone those that would produce an interior maximum at a regime transition.*"

- l. 298: is it clear here why the transition altitude is the altitude of the ozone maximum?

  This is a good question, the answer to which can be inferred from the paper. Under gray radiation, ozone increases monotonically towards the surface in the O-damped regime, so the interior maximum cannot be within the O-damped regime. So the maximum of the O-damped regime must be at the lower boundary, i.e., the regime transition. Now, rather than occurring at the regime transition, can the interior maximum occur within the $O_3$-damped regime? This is possible in certain limits, that correspond to the strongly $O_3$-damped case where the ozone maximum is explained by the source-controlled paradigm. In this case, our analytical expressions would still apply to the ozone layer, but the interior maximum would occur in the $O_3$-damped regime. Very strong damping of $O_3$ is required to shift the maximum into the $O_3$-damped regime, which is why we have written, "*In today's atmosphere (and across a quite wide parameter regime), this transition altitude is also the altitude of peak $[O_3]$.*

- l. 314: the surface is not a region where these theories should be applied.

  Indeed, we have revised this sentence to, *"Absent a transition to an $O_3$-damped regime, the O-damped ozone layer would therefore increase all the way down and have no interior maximum.*

- l.430,431: I am confused here: the red line in Fig. 5 is discussed (source/sink paradigm) — is does not provide a good estimate for the ozone maximum. However the Chapman+2 model (magenta line) does. So why are we concerned about the red line if the Chapman+2 model seems appropriate?

  Our goal in Figure 5 is to further understand the Chapman+2 model. The Chapman+2 model has two dominant regimes which together lead to the interior maximum. In Figure 5, we are seeking to understand the ozone maximum that would result if only one of those regimes was dominant everywhere. The red line shows the ozone profile that would result in the source/sink competition paradigm. The blue line shows the ozone profile that would result in the source-controlled paradigm. These regimes are being studied in order to understand the Chapman+2 model, whose interior maximum is appropriate but can be beneficially interpreted with the help of studying these limiting cases.

- l. 436: 'generalized destruction' is not clear.

  In the process of clarifying the description of why the Chapman cycle predicts the right interior maximum for the wrong reason, we have removed this confusing phrase.

- l. 473: change $*$ to $\cdot$

  Corrected.

- l. 476: why are the absorption coefficients not taken from the most recent kinetic recommendation (Burkholder et al., 2020)?

  Our results are not significantly dependent when choosing from among the standard absorption coefficient datasets of the past 50 years. The recommended absorption coefficients for $O_3$ that we used from the Sander et al. (2010) have not changed in the most recent JPL recommendation of Burkholder et al. (2019), so we will revise to cite the more recent dataset. The recommended absorption coefficients for $O_2$ combine those listed in Burkholder et al. (2019) for $\lambda > 205$ nm and Kockarts (1976) for $\lambda < 205$ nm. We have compared our version using Ackerman (1971) to this recommended set from Burkholder et al. (2019), and find that the differences are minor, with the ozone maximum changing by 0.4 km in response to the change in $\sigma_{O_2}$. The resulting change in the ozone profile is shown in Figure 3 of this Response to Reviewers. This is within our tolerance of approximating the ozone maximum to O(1 km). That said, in the interest of using the latest absorption coefficients, we have adopted the Burkholder et al. (2019) coefficients for $\sigma_{O_2}$ throughout the

paper. All figures and numbers have been recalculated with the new profile, leading to at most minor updates. The most significant change as a result of this update is that the predicted peak of the source/sink competition paradigm (i.e., the O-damped regime), which occurs outside its range of applicability, has shifted from 17 km to 15 km, a small change in the direction of strengthening our argument that the interior maximum cannot be understood from the source/sink competition paradigm on its own.

[Figure]

Figure 3: Sensitivity to absorption coefficient datasets for $O_2$, comparing the coefficients from Ackerman (1971) (first submitted version) with those from Burkholder et al. (2019) (now used in revised version).

- l. 477: Why is it not possible to approximately take the atmospheric temperature profile into account when calculating temperature dependent kinetic parameters?

  It is not impossible to take an average temperature as a function of altitude and use that to make the collisional reaction rate coefficients, which are generally temperature-dependent, also functions of altitude. Figure 4 of this Response to Reviewers shows the result. This added complexity leads to only an 11% change in the magnitude of the ozone maximum, which shifts up by 0.5 km. Both of these sensitivities are within our tolerance of reproducing the interior maximum of ozone to O(1 km) while providing readers with a minimal theory for the interior maximum of ozone. No previous work has suggested that the temperature structure of the stratosphere is responsible for the interior maximum of ozone, and Figure 4 vindicates this lack of emphasis. Because we want to positively imply through our methodology that the interior maximum of ozone does not depend significantly on vertical temperature variations, we have retained our isothermal methodology, which is subsequently necessary when deriving an analytical expression for the ozone profile.

[Figure]

Figure 4: Sensitivity of tropical ozone to the assumption of an isothermal atmosphere compared to using tropically-averaged (30S-30N) temperature from SWOOSH (1984-2023), which is based on MERRA-2 (Gelaro et al., 2017).

- Fig. C1: I cannot see the magenta line mentioned in the caption in this figure.

  The legend has been corrected to referring to the black line.

- l. 614: 'The atmospheric environment' is listed here twice.

  Corrected.

**2.3 References**

Brasseur, G. P.: The Ozone layer: From Discovery to Recovery, American Meteorological Society, 2019.

Burkholder, J. B., Sander, S. P., Abbatt, J. P. D., Barker, J. R., Cappa, C., Crounse, J. D., Dibble, T. S., Huie, R. E., Kolb, C. E., Kurylo, M. J., Orkin, V. L., Per- cical, C. J., Wilmouth, D. M., and Wine, P. H.: Chemical kinetics and pho- tochemical data for use in atmospheric studies, Evaluation Number 19, JPL Publication 19-5, URL http://jpldataeval.jpl.nasa.gov, 2020.

Chapman, S.: A theory of upper atmospheric ozone, Mem. Roy. Soc., 3, 103-109, 1930.

Crutzen, P. J., Groo, J.-U., Bru?hl, C., Mu?ller, R., and Russell III, J. M.: A Reevaluation of the ozone budget with HALOE UARS data: No evidence for the ozone deficit, Science, 268, 705-708, 1995.

Davis, S. M., Rosenlof, K. H., Hassler, B., Hurst, D. F., Read, W. G., Vo?mel, H., Selkirk, H., Fujiwara, M., and Damadeo, R.: The Stratospheric Water and Ozone Satellite Homogenized (SWOOSH) database: a long-term database for climate studies, Earth System Science Data, 8, 461-490, https://doi.org/ 10.5194/essd-8-461-2016, 2016.

Diallo, M., Riese, M., Birner, T., Konopka, P., Mu?ller, R., Hegglin, M. I., Santee, M. L., Baldwin, M., Legras, B., and Ploeger, F.: Response of stratospheric water vapor and ozone to the unusual timing of El Nin?o and the QBO disrup- tion in 2015-2016, Atmos. Chem. Phys., 18, 13055-13073, https://doi.org/ 10.5194/acp-18-13055-2018, 2018.

Dobson, G. M. B.: Exploring the atmosphere, Oxford University Press, 1963.

Gelaro, R., McCarty, W., Sua?rez, M. J., Todling, R., Molod, A., Takacs, L., Randles, C. A., Darmenov, A., Bosilovich, M. G., Reichle, R., Wargan, K., Coy, L., Cullather, R., Draper, C., Akella, S., Buchard, V., Conaty, A., da Silva, A. M., Gu, W., Kim, G.-K., Koster, R., Lucchesi, R., Merkova, D., Nielsen, J. E., Partyka, G., Pawson, S., Putman, W., Rienecker, M., Schubert, S. D., Sienkiewicz, M., and Zhao, B.: The Modern-Era Retrospective Analysis for Research and Applications, Version 2 (MERRA-2), jci, 30, 5419-5454, 2017.

Han, Y., Tian, W., Chipperfield, M. P., Zhang, J., Wang, F., Sang, W., Luo, J., Feng, W., Chrysanthou, A., and Tian, H.: Attribution of the Hemispheric Asymmetries in Trends of Stratospheric Trace Gases Inferred From Microwave Limb Sounder (MLS) Measurements, J. Geophys. Res., 124, 6283-6293, https://doi.org/https://doi.org/10.1029/2018JD029723, 2019.

Hoffmann, L. and Spang, R.: An assessment of tropopause characteristics of the ERA5 and ERA-Interim meteorological reanalyses, Atmos. Chem. Phys., 22, 4019-4046, https://doi.org/10.5194/acp-22-4019-2022, 2022.

Nivano, M., Yamazaki, K., and Shiotani, M.: Seasonal and QBO variations of as- cent rate in the tropical lower stratosphere as inferred from UARS HALOE trace gas data, J. Geophys. Res., 108, 4794, https://doi.org/10.1029/2003JD003871, 2003.

Portmann, R. W., Daniel, J. S., and Ravishankara, A. R.: Stratospheric ozone depletion due to nitrous oxide: influences of other gases, Phil. Trans. R. Soc. B, 367, 1256-1264, https://doi.org/10.1098/rstb.2011.0377, 2012.

Waters, J. W., Froidevaux, L., Harwood, R. S., Jarnot, R. F., Pickett, H. M., Read, W. G., Siegel, P. H., Cofield, R. E., Filipiak, M. J., Flower, D. A., Holden, J. R., Lau, G. K., Livesey, N. J., Manney, G. L., Pumphrey, H. C., Santee, M. L., Wu, D. L., Cuddy, D. T., Lay, R. R., Loo, M. S., Perun, V. S., Schwartz, M. J., Stek, P. C., Thurstans, R. P., Boyles, M. A., Chandra, S., Chavez, M. C., Chen, G.-S., Chudasama, B. V., Dodge, R., Fuller, R. A., Girard, M. A., Jiang, J. H., Jiang, Y., Knosp, B. W., LaBelle, R. C., Lam, J. C., Lee, K. A., Miller, D., Oswald, J. E., Patel, N. C., Pukala, D. M., Quintero, O., Scaff, D.

M., Snyder, W. V., Tope, M. C., Wagner, P. A., and Walch, M. J.: The Earth Observing Sys- tem Microwave Limb Sounder (EOS MLS) on the Aura satellite, IEEE Trans. Geosci. Remote Sens., 44, 1106-1121, 2006.

**References**

Ackerman, M., 1971: Ultraviolet Solar Radiation Related to Mesospheric Processes. Springer, Dordrecht, 149–159, doi:10.1007/978-94-010-3114-1_11.

Andrews, D. G., J. R. Holton, and C. B. Leovy, 1987: *Middle Atmosphere Dynamics.* Academic Press.

Bates, D. R., and M. Nicolet, 1950: The photochemistry of atmospheric water vapor. *Journal of Geophysical Research (1896-1977)*, **55 (3)**, 301–327, doi:10.1029/JZ055i003p00301.

Brasseur, G. P., and S. Solomon, 2005: *Aeronomy of the Middle Atmosphere: Chemistry and Physics of the Stratosphere and Mesosphere.* Springer, Dordrecht, Netherlands.

Burkholder, J. B., and Coauthors, 2019: Chemical Kinetics and Photochemical Data for Use in Atmospheric Studies. Tech. rep., Jet Propulsion Laboratory, Pasadena, CA.

Crutzen, P. J., 1970: The influence of nitrogen oxides on the atmospheric ozone content. *Quarterly Journal of the Royal Meteorological Society*, **96 (408)**, 320–325, doi:10.1002/qj.49709640815.

Davis, S. M., and Coauthors, 2016: The Stratospheric Water and Ozone Satellite Homogenized (SWOOSH) database: A long-term database for climate studies. *Earth System Science Data*, **8 (2)**, 461–490, doi:10.5194/essd-8-461-2016.

Dobson, G. M. B., 1956: Origin and Distribution of the Polyatomic Molecules in the Atmosphere. *Proceedings of the Royal Society A: Mathematical, Physical and Engineering Sciences*, **236 (1205)**, 187–193, doi:10.1098/rspa.1956.0127.

Fels, S. B., 1982: A Parameterization of Scale-Dependent Radiative Damping Rates in the Middle Atmosphere.

Garcia, R. R., and S. Solomon, 1985: The effect of breaking gravity waves on the dynamics and chemical composition of the mesosphere and lower thermosphere. *Journal of Geophysical Research*, **90 (D2)**, 3850, doi:10.1029/JD090iD02p03850.

Gelaro, R., and Coauthors, 2017: The Modern-Era Retrospective Analysis for Research and Applications, Version 2 (MERRA-2). *Journal of Climate*, **30 (14)**, 5419–5454, doi:10.1175/JCLI-D-16-0758.1.

Gill, A. E., 1980: Some simple solutions for heat-induced tropical circulation. *Quarterly Journal of the Royal Meteorological Society*, **106 (449)**, 447–462, doi:10.1002/qj.49710644905.

Held, I. M., and M. J. Suarez, 1994: A Proposal for the Intercomparison of the Dynamical Cores of Atmospheric General Circulation Models. *Bulletin of the American Meteorological Society*, **75 (10)**, 1825–1830, doi:10.1175/1520-0477(1994)075⟨1825:APFTIO⟩2.0.CO;2.

Jacob, D., 1999: *Introduction to Atmospheric Chemistry*. Princeton University Press.

Kockarts, G., 1976: Absorption and photodissociation in the Schumann-Runge bands of molecular oxygen in the terrestrial atmosphere. *Planetary and Space Science*, **24 (6)**, 589–604, doi:10.1016/0032-0633(76)90137-9.

Match, A., and E. P. Gerber, 2022: Tropospheric Expansion Under Global Warming Reduces Tropical Lower Stratospheric Ozone. *Geophysical Research Letters*, **49 (19)**, e2022GL099 463, doi:10.1029/2022GL099463.

Neu, J. L., and R. A. Plumb, 1999: Age of air in a "leaky pipe" model of stratospheric transport. *Journal of Geophysical Research*, **104 (D16)**, 19 243, doi:10.1029/1999JD900251.

Perliski, L. M., S. Solomon, and J. London, 1989: On the interpretation of seasonal variations of stratospheric ozone. *Planetary and Space Science*, **37 (12)**, 1527–1538, doi:10.1016/0032-0633(89)90143-8.

Portmann, R. W., J. S. Daniel, and A. R. Ravishankara, 2012: Stratospheric ozone depletion due to nitrous oxide: Influences of other gases. *Philosophical Transactions of the Royal Society B: Biological Sciences*, **367 (1593)**, 1256–1264, doi:10.1098/rstb.2011.0377.

Romps, D. M., 2014: Rayleigh Damping in the Free Troposphere. *Journal of Atmospheric Sciences*, **71**, 553–565, doi:10.1175/JAS-D-13-062.1.

Sander, S. P., and Coauthors, 2010: Chemical Kinetics and Photochemical Data for Use in Atmospheric Studies, Evaluation No. 17. Tech. Rep. 10-6, Jet Propulsion Laboratory, Pasadena, CA.

Stolarski, R. S., D. W. Waugh, L. Wang, L. D. Oman, A. R. Douglass, and P. A. Newman, 2014: Seasonal variation of ozone in the tropical lower stratosphere: Southern tropics are different from northern tropics. *Journal of Geophysical Research: Atmospheres*, **119 (10)**, 6196–6206, doi:10.1002/2013JD021294.

---

## Author Response (AR2)

**Response to technical corrections: "Protection without poison: why tropical ozone maximizes in the interior of the atmosphere" (egusphere-2024-1552)**

Aaron Match, Edwin P. Gerber, and Stephan Fueglistaler

December 4, 2024

We have made brief technical corrections for the revised version of our paper, and the reviewer response is included below. We again thank the editor and reviewers for their valuable feedback, with our responses noted in blue.

In addition to these technical corrections requested by the reviewer, we have additionally proof-read the paper and made a small number of minor revisions of our own, primarily of a grammatical or stylistic nature. Also, we now cite four textbooks for each textbook paradigm in the Introduction, totaling 8 textbooks, a change that reflects a more strict criterion for inferring whether the textbook supports a particular paradigm.

**1 Reviewer comments**

I see that this paper has seen a lot of work in response to the first set of reviews; a new Figure (7) was added and the figures and numbers in this paper have been recalculated based on the most recent kinetic information. Congratulations to accomplishing this work. Further, many minor modifications were also made.

I suggest accepting this paper for ACP in its present form.

We thank the reviewer again for their careful review of this work, and their constructive feedback.

I have a very few comments (see below) that the authors might want to consider when producing a revised form of this paper.

\* I like the idea of using SWOOSH in place of MERRA-2 (and I agree the changes are minor). But I think the paper is clearer in this way.

We agree that SWOOSH has improved the paper.

\* I also like the idea removing the troposphere from the figures.

We agree that removing the troposphere has improved the paper.

\* Oy is defined in Brasseur and Solomon (2005). Indeed, it is not uncommon (for certain applications) to have more species contributing to Ox than just O3 and O. However,

I suggest adding the Oy fedinition to this paper, so that people do not have to look up the Oy definition in the book by Brasseur and Solomon (2005).

Per this request, at the location of our first mention of $O_y$, we have added a footnote that includes its chemical definition from Brasseur and Solomon (2005). The footnote is as follows on page 8: *On their page 414, $O_y$ is defined as $O(^3P) + O(^1D) + O_3 + NO_2 + 2NO_3 + HNO_3 + HO_2NO_2 + 2N_2O_5 + ClO + 2Cl_2O_2 + 2OClO + 2ClONO_2 + BrO + 2BrONO_2$.*

* regarding the Cariolle coefficients, I agree with you that they should nor be published with out his explicit consent. Perhaps there is a chance that D. Cariolle eventually agrees to his values being published.

We agree.

* Regarding Dobson: While he measured total ozone in many locations globally, for some time he estimated the peak of the ozone layer to be above 30 km. These estimates can be found in his old papers. Of course his book was later.

Thanks for pointing out this historical note. We are curious to read about how Dobson's understanding evolved over the years. We now cite Dobson (1963) as an example of a source/sink competition paradigm explanation.

* You mention in your reply that it is not impossible to consider temperature as a function of altitude. Perhaps this point could briefly be mentioned in the paper (rather than making the assumption "isothermal" right away.

Our introduction to the simplifying approximations of the Chapman+2 model now clarifies more explicitly that some of the approximations, namely that of an isothermal atmosphere, can be relaxed in our modeling framework, although we still employ them to support the derivation of the analytical solutions: *Agreement between the Chapman+2 model and observations is imperfect, which is unsurprising given that this work employs many simplifying approximations. Many of these approximations are required to subsequently derive an explicit analytical expression to the Chapman+2 model ozone profile. For example, we will present results for overhead sun impinging on an isothermal atmosphere, although our model can also be run at other solar zenith angles or with vertically-varying temperature. We have also approximated transport and catalytic cycles as a linear damping, used globally-averaged catalytic profiles, and neglected optical scattering. Despite these approximations, the Chapman+2 model produces a reasonable fit to the observed profile, and will be considered to produce a credible interior maximum of ozone. The remainder of the paper seeks to explain why the Chapman+2 model produces an interior maximum.*

**References**

Dobson, G. M. B., 1963: *Exploring the Atmosphere, by G.M.B. Dobson.* Clarendon Press, Oxford.